

# Interactions of Atmospheric Gases and Aerosols with the Monsoon Dynamics over the Sudano-Guinean region during AMMA

Adrien DEROUBAIX[1,2], Cyrille FLAMANT[2], Laurent MENUT[1], Guillaume SIOUR[3], Sylvain MAILLER[1], Solène TURQUETY[1], Régis BRIANT[1], Dmitry KHVOROSTYANOV[1], and Suzanne CRUMEYROLLE[4]

[1]LMD/IPSL, École Polytechnique, Université Paris Saclay, ENS, IPSL Research University; Sorbonne Universités, UPMC Univ Paris 06, CNRS, Palaiseau, France
[2]LATMOS/IPSL, UPMC, Sorbonne Universités, CNRS & UVSQ, Paris, France
[3]LISA/IPSL, Universités Paris Est Créteil & Paris Diderot, Créteil, France
[4]LOA, Université Lille 1 Sciences et Technologies, Villeneuve d'Ascq, France

*Correspondence to:* Adrien Deroubaix, adrien.deroubaix@lmd.polytechnique.fr

**Abstract.** Carbon monoxide, CO, and fine atmospheric particulate matter, $PM_{2.5}$, are analyzed over the Guinean gulf coastal region using the WRF-CHIMERE modeling system and observations during the beginning of the monsoon 2006 (from May to July), corresponding to the Africa Multidisciplinary Monsoon Analysis (AMMA) campaign period.

Along the Guinean gulf coast, the contribution of long-range pollution transport to CO or $PM_{2.5}$ concentrations is important.

For $PM_{2.5}$, desert dust decreases from $\approx 38\ \%$ in May to $\approx 5\ \%$ in July; biomass burning aerosol from Central Africa increases from $\approx 10\ \%$ in May to $\approx 52\ \%$ in July. The anthropogenic contribution is $\approx 30\ \%$ for CO and $\approx 10\ \%$ for $PM_{2.5}$.

When focusing only on anthropogenic pollution, frequent northward transport events from the coast to the Sahel are associated with periods of low wind and no precipitation. In June, anthropogenic $PM_{2.5}$ and CO concentrations are higher than in May or July over the Guinean coastal region. Over the Sahel, air masses dynamics concentrate pollutants emitted locally and remotely at the coast due to a meridional atmospheric cell.

Refining the analysis on the period 8 - 15 June, anthropogenic pollutants emitted along the coastline are exported toward the North especially at the beginning of the night (18 UTC to 00 UTC) with the establishment of the nocturnal low level jet. Plumes originating from different cities overlay for some hours at the coast, leading to high pollution level, because of specific disturbed meteorological conditions.

## 1 Introduction

The interactions between air pollution and climate in megacities is a challenging field of research (Baklanov et al., 2016). In the countries of the Guinean Gulf, the population has been growing rapidly during the last decades, accompanied by economic development. Parallel to industrialization, air pollution is increasing without any governmental control (Zhu, 2012).

During the dry season (*i.e.* November - April), high ozone concentrations and smog are observed over megacities such as Lagos or Cotonou when the Harmattan easterly wind is weak (Marais et al., 2014; Minga et al., 2009). During the wet



season (*i.e.* May - October), the West African Monsoon (WAM) wind carries the pollutants northward, and local convective precipitations wash out the atmosphere. Two precipitation periods occur over the Guinean Gulf coastal region in April-May and August-September. Between these two periods, the wind coming from the South is predominant (Janicot et al., 2008).

There are various air pollution sources in the Guinean Gulf coastal region during the WAM. Sea salt aerosols are transported
in the marine boundary layer, and mineral dust aerosols are transported in the Saharan Air Layer (SAL) above the monsoon air (Lafore et al., 2011). Biogenic components are emitted by tropical forests (Reeves et al., 2010), and the urban air pollution in megacities (Liousse et al., 2014). In addition, pollutants resulting from incomplete combustion such as carbon monoxide and black carbon particles coming from the Southern hemisphere due to biomass burning reach the Guinean coast in June (Mari et al., 2007). Biomass burning plumes observations have shown high ozone ($\geq$ 60 ppb at 700 hPa) and carbon monoxide
concentration ($\geq$ 200 ppb at 700 hPa) (Sauvage et al., 2004; Mari et al., 2011).

In Nigeria, Akeredolu (1989) have listed the different sources of particle loading: biomass burning (31.7 %), fugitive dust from roads (29.1 %), fuel wood burning (21.3 %), Harmattan dust (13.8 %), solid waste incineration (2.1 %), stationary sources (1.6 %), automobile exhaust lead (0.2 %) and gas flares (0.1 %). Since the 1990's, natural pollutions from desert dust and vegetation fires remains important (Mari et al., 2011; Haywood et al., 2008). However, anthropogenic pollution has increased:
there is more local fuel-wood burning for stoves and more traffic (Liousse et al., 2010; Hadji et al., 2012; Liousse et al., 2014) with more two-wheel vehicles which are suspected to quickly worsen the air quality, partly due to the very poor fuel quality used (Ndoke and Jimoh, 2005; Assamoi and Liousse, 2010); the economic growth over the region drives up emissions by industries including gas flaring (Asuoha and Osu, 2015).

All studies of air quality monitoring have shown that the outdoor air quality standards (*i.e.* threshold concentrations) are
largely exceeded. These thresholds are for CO: 35 ppm for 1 h and 9 ppm for 8 h exposure; and for $PM_{2.5}$: 10 $\mu$g.m$^{-3}$ annual mean, 25 $\mu$g.m$^{-3}$ for 24-hour mean. For instance, in April 1993, Baumbach et al. (1995) have measured in Lagos (Nigeria) high levels of carbon monoxide, CO, (up to 10 ppm, measured close to high traffic road) and total particulate matter (up to 200 $\mu$g.m$^{-3}$). In Accra (Ghana), Dionisio et al. (2010) have measured $PM_{2.5}$ up to 200 $\mu$g.m$^{-3}$ in a polluted street by wood stoves, heavy traffic and trash burning. In Ouagadougou (Burkina Faso) $PM_{2.5}$ reach 164 $\mu$g.m$^{-3}$ (Boman et al., 2009) and CO
concentration measured in-traffic frequently exceed all World Health Organization (WHO) guidelines (Lindén et al., 2008).

The health impact of such air pollution is expected to be high and to increase without any specific emission regulation (Lindén et al., 2012). It is therefore important to gain a better understanding of the pollutants'emissions and transport in West Africa. All these results have highlighted the high level of pollution in megacities affecting remote places. However, there is no continuous air quality monitoring in West Africa, so existing studies are focused on local scale, short time period, and few
pollutants.

Several observation campaigns have been dedicated to WAM since the last decade, notably the Africa Multidisciplinary Monsoon Analysis (AMMA) which was the first international program started to improve our knowledge of all aspects of the WAM (Redelsperger et al., 2006). WAM modeling made progresses, however the Guinean Gulf coastal region is challenging to model because of the complex land-sea-atmosphere interactions.





Along the coastline, there are several atmospheric cells acting at different scales. The diurnal cycle of the land-sea breeze occurs at a local scale (few kilometers). During the day, surface wind is linked with convection within the boundary layer, while at night there is the formation of the Nocturnal Low Level Jet (NLLJ) in response to the daily deep convection activity (Parker et al., 2005). At a regional scale (few hundred kilometers), the monsoon wind from the South meets the Harmattan wind from the North, forming the Inter Tropical Discontinuity at the ground level (Flamant et al., 2007; Cuesta et al., 2009; Karam et al., 2009; Pospichal et al., 2010), leading to a complex vertical structure (Haywood et al., 2008; Lafore et al., 2011). Between these two scales, an additional meridional atmospheric cell is suspected in the low atmosphere enhancing convergence at the coast (Leduc-Leballeur et al., 2013), which results from a gradient of wind speed due to the meridional gradient of sea surface temperature (de Coëtlogon et al., 2014). The recent research program "Dynamics-Aerosol-Chemistry-Cloud Interactions in West Africa program" (DACCIWA) has been dedicated to the study of land-sea-atmosphere interactions in West Africa. It will contribute to the development of the next generation of accurate models to forecast weather and pollution in southern West Africa (Knippertz et al., 2015).

This article is dedicated to the pollutants transport over the Guinean Gulf coastal region and focuses on two major pollutant concentrations: Carbon monoxide and Particulate Matter with an aerodynamic diameter $D_p < 2.5$ $\mu$m (CO and $PM_{2.5}$ hereafter), which have both an detrimental impact on health (Lelieveld et al., 2015). The scientific questions addressed in this work are:

- *What is the relative contribution of long-range transported and locally emitted pollutants in the surface concentrations from the Guinean Gulf to the Sahel?*

- *What is the impact of meridional atmospheric cells on the transport of pollutants emitted from coastal megacities?*

The pollution patterns are analyzed during the 2006 AMMA period using several observational data sets in combination with numerical simulations of the meteorology as well as the aerosol-gas chemistry and transport presented in section 2. Section 3 presents the main spatial and temporal patterns over the Sudano-Guinean region of the AMMA study case. Section 4 analyzes the anthropogenic pollution from the coast to the Sahel. Section 5 refines spatially the analysis on the coastal dynamics and pollution transport. Section 6 focuses on specific study cases. Conclusions and perspectives are given in section 7.

## 2 Weather-Pollution modeling configuration

The modeling analysis was performed using the Weather Research and Forecasting (WRF) model for the meteorological fields, which drives the CHIMERE model for the gaseous and particulate species concentrations. Two nested geographical domains are defined: a continental one to take into account remote sources and long-range transport from the Mediterranean sea to the tropic of Capricorn (27°S to 44°N; 38°W to 47°E); and a regional one, centered on the Guinean Gulf (1°N to 20°N; 23°W to 17°E). WRF and CHIMERE models work on the same two horizontal grids. The time period simulated is April to end of July 2006, including a one month spin-up.



## 2.1 Meteorological fields with the WRF model

The meteorological variables are modeled with the regional non-hydrostatic WRF model (version 3.7.1) presented by Ska-marock and Klemp (2008). The continental domain has a constant horizontal resolution of 60 km × 60 km, and of 20 km × 20 km for the regional one, both with 32 vertical levels from the surface to 50 hPa. We use a 2-way nesting with the WRF model.

The global meteorological fields are taken from the US Global Forecast System produced by the National Center for Environmental Prediction. It is read and hourly interpolated by WRF using low frequency spectral nudging above the PBL in order to enable the PBL variability to be resolved by WRF (von Storch et al., 2000). We followed the recommendations of Flaounas et al. (2010, 2011) to configure the convection and planetary boundary layer schemes, which have optimized a better model set-up for the entire 2006 WAM, especially for the meridional gradient of temperature and the low level circulation.

The Single Moment-6 class microphysics scheme (WSM6) is used allowing for mixed phase processes suitable for high resolution simulations (Hong and Lim, 2006). Li et al. (2015) have shown that WAM precipitation patterns are very sensitive to the radiation scheme, and the most realistic patterns were obtained with the Rapid Radiative Transfer Model for General Circulation Models (RRTMG) with the Monte-Carlo Independent Column Approximation (McICA) method of random cloud overlap from Mlawer et al. (1997). The planetary boundary layer physics are computed using the Yonsei University scheme

(Hong et al., 2006). The cumulus parametrization used is the ensemble Grell-Dévényi scheme, as Crétat and Pohl (2012) have shown that internal variability is much larger with the Kain-Fritsch scheme than for the Grell-Dévényi scheme at the seasonal, intra-seasonal, and daily time scales, and from the regional to the local (grid point) spatial scales. The surface layer scheme is based on Monin-Obukhov with Carslon-Boland viscous sub-layer. The surface physics are calculated using the 'Noah' Land Surface Model) scheme with four soil temperatures and moisture layers Ek et al. (2003).

## 2.2 Chemistry-Transport with the CHIMERE model

CHIMERE is a regional chemistry-transport model (version 2017), fully described in Menut et al. (2013a); Mailler et al. (2016). The CHIMERE model has previously been used over the AMMA observation period but only dust aerosols were modeled (Schmechtig et al., 2011; Menut et al., 2009). In this study, all important sources are included (anthropogenic, biogenic, mineral dust, sea salt and biomass burning). The 32 vertical levels of the WRF model are projected on the 20 levels for CHIMERE

from the surface to 200 hPa. We use a 1-way nesting with the CHIMERE model.

The anthropogenic emissions are estimated using the HTAP v2 (Hemispheric Transport of Air Pollution) annual totals for the year 2010 by the EDGAR Team, using inventories based on MICS-Asia, EPA-US/Canada and TNO databases (available at *http://edgar.jrc.ec.europa.eu/htap_v2*). Figure 1 presents the anthropogenic PM and CO emissions over the regional domain and the Cotonou-Niamey meridional transect used for the analysis in the next sections, defined in longitude $\lambda = 2°$ East to $3°$

East, and in latitude $\phi = 1°$ North to $19°$ North.

Taking into account vegetation fires emission fluxes is of primary importance to simulate West African pollution (Giglio et al., 2006). This is achieved using the APIFLAME model (Turquety et al., 2013), which estimates aerosols and chemical species emissions produced by vegetation fires. Since the incomplete combustion is both included in anthropogenic inventories





(local urban burning) and fires emissions inventories (biomass burning of forests), the simulation was designed to split these two parts.

Biogenic emissions are calculated using the MEGAN emissions scheme (Guenther et al., 2006). The mineral dust sources are obtained using a new global soil and surface datasets made from a satellite-derived aeolian roughness length data with a 6
km spatial resolution GARLAP (Global Aeolian Roughness Lengths from ASCAT and PARASOL) (Menut et al., 2013b).

Bessagnet et al. (2004) described the calculation of gaseous species in the MELCHIOR-2 (reduced) scheme and the aerosol scheme, which takes into account species such as sulphate, nitrate, ammonium, primary organic matter (POM) and elemental carbon (EC), secondary organic aerosols (SOA), sea salt, dust and water. All aerosols are represented using ten bins, from 40 nm to 40 $\mu$m in diameter. Their life cycle is fully represented with emission, transport, chemistry and deposition (wet and dry).
The top and lateral boundary conditions are driven by LMDZ-INCA for aerosols and chemical species (Folberth et al., 2006). It is also possible to release gaseous or particulate atmospheric tracers, which is a powerful tool to analyze the pollution patterns.

## 3 Temporal variability from May to July 2006

In this section, we first analyze and quantify the temporal variability of the pollutants concentrations modeled in the urbanized areas along the Guinean Gulf coast during the whole AMMA-SOP1 period (Redelsperger et al., 2006). First, the precipitation
regimes are analyzed using Hovmoller diagrams. Second, AERONET surface stations data are used to quantify the three-months variability of the Aerosol Optical Depth (AOD). Finally, the relative contribution of several sources are quantified using the model. The two last points focus on three locations: Cotonou (Benin), Djougou (Benin) and Niamey (Niger), which are representative of locations under several influences (mineral dust, anthropogenic pollution and vegetation fires).

### 3.1 Precipitations Patterns

During this period, the precipitation location and rate will play a crucial role on the modeled surface $PM_{2.5}$ concentrations. As a validation for this variable, the methodology of Flaounas et al. (2010) is used: precipitation rates are averaged between 8.5°W and 8.5°E. Day-to-day variability is smoothed by applying a moving average of $\pm 2$ days. Figure 2 is directly comparable to the Flaounas et al. (2010) study using the same period and averaged region. Results show that the modeled precipitation patterns are in good agreement with the two satellite observations (TRMM and GPCP) presented in their study.

The WAM is due to the sea surface temperature decreases, which forms a cold tongue, and over the Sahara, a low thermal pressure system appears called the Saharan Heat Low (Lafore et al., 2011). The temperature gradient between the sea and the Sahara allows the monsoon system to progress inland reaching the Sahara in July (Hall and Peyrillé, 2006; Lavaysse et al., 2009). We can notice on Figure 2 that the monsoon progression to the North is not linear. Two jumps are observed due to the complex sea-land-atmosphere interactions constituted of two steps: the monsoon 'pre-onset' occurs end of May when the main
precipitation area associated to the Inter Tropical Convergence Zone (ITCZ) located at 2°N moves to 5°N (Sultan and Janicot, 2000; Sultan et al., 2003; Sultan and Janicot, 2003); and the monsoon 'onset' is another abrupt shift happening end of June





when the main precipitation area moves from 5° to 10°N (Janicot et al., 2008). The precipitation is associated with large scale squall-lines, creating Mesoscale Convective Systems (MCS) moving Westward (Hall and Peyrillé, 2006).

The meteorological simulation reproduces changes of the main precipitation area at the end of May (*i.e.* the 'pre-onset') and at the beginning of July (*i.e.* the 'onset'). For these dates, simulated precipitations match very well AMMA observations which have shown that the 2006 monsoon onset date was the 10 July. In 2006, the monsoon onset occurred with a 10-day delay compared to its climatological date, *i.e.* 24 June with a standard deviation of 8 days over the period 1968-2005 (Janicot et al., 2008). Thus, three periods could be defined: before 'pre-onset' (in May), between 'pre-onset' and onset (in June), after onset (in July).

## 3.2 Meridional aerosols content

In our studied region, surface concentrations in the cities are affected by several contributions. In addition to local emission, cities may be strongly impacted by biomass burning transported from the Central Africa (Mari et al., 2007), or by mineral dust transported from Sahara (Flamant et al., 2009).

The modeled daytime Aerosol Optical Depth (AOD) and Angström exponent are compared to observations from the AERONET network (Holben et al., 1998), available at (*aeronet.gsfc.nasa.gov*). From the daily AERONET level-2 measurements AERONET-AOD at 440nm and Angström exponent 440-870, AERONET-AOD is calculated at 600 nm based on the Angström law. A spatial bilinear interpolation of the model outputs is performed at the station location.

Two AERONET stations are located close to the meridional transect studied (Figure 1): Banizoumbou (13.5°N, 2.1°E) in the suburb of Niamey in Niger, and Djougou in Benin (9.7°N, 1.6°E) North of the strongly urbanized areas around Cotonou. Comparisons are presented in Figure 3.

There are two important events of coarse particles recorded at both sites, associated with a low Angström exponent (*i.e.* Angström exponent lower than 0.5 as in Ogunjobi et al. (2008)) and AOD greater than 1, between 13-14 May and between 10-13 June. The model captures the magnitude of these large scale dust events. During the studied period, the events of coarse particles are well reproduced (high or moderate AOD are generally associated with a low Angström exponent). There is an increase of the Angström exponent, *i.e.* fine particles over the period, which is well reproduced by the model. Frequent fine aerosol events (high Angström exponent) have been monitored corresponding to low or moderate AOD which are partially captured by the model.

The addition of the biomass burning emission lead to an important plume of gas and aerosols reaching the Guinean gulf in June. Modeled AOD are well in the range with biomass emission (bias is reduced) but the variability is not captured. The Angström exponent is associated in May with coarse particles (Angström exponent about 0.2), in June with fine/coarse mixture of particles (about 0.5), in July with a fine particles (about 0.8). The model is able to reproduce this increase of the Angström exponent, which suggests an aerosol origin transition, from a period dominated by desert dust to a period of fine particles which could be urban or/and biomass burning pollution from the South.



### 3.3 Meridional aerosols and gases concentrations

In this section, firstly the modeled CO and $PM_{2.5}$ concentrations are compared to observations collected during the AMMA campaign. Secondly, the different contribution of the pollution sources is analyzed from the modeled concentrations at the three studied sites.

### 3.3.1 Airborne observations

Two flights made during a 'North-South land-atmosphere-ocean interaction' mission plans have been conducted over the Cotonou-Niamey meridional transect on 13 and 14 June 2006. These two days correspond to disturbed dynamics of the WAM due to a MCS. It developed in the vicinity of the Jos plateau in the North of Nigeria around 16 UTC, moving westward to the center of Ghana, which have already been described (*e.g.* (Flamant et al., 2009) and (Crumeyrolle et al., 2011)). Moreover, the MCS interacts with the dust layer coming from the Sahara (especially from the Bodele depression), changing the dust load and

vertical distribution over Benin and Niger. Associated with subsidence in the wake of the MCS, there is a lowering of the dust layer height (Flamant et al., 2007).

Modeled CO and $PM_{2.5}$ concentrations are compared to aircraft measurements performed onboard the ATR-42 aircraft (with PCASP instrument for PM), which have been averaged at a 2-minute time step. The modeled values are interpolated along the aircraft trajectories, in time between the two closest modeled hourly outputs, and vertically between the two closest model

vertical levels and horizontally with a bilinear interpolation. For two flights (13 July in the morning from Niamey to Cotonou, and 14 July in the afternoon from Cotonou to Niamey), Table 1 presents modeled and observed mean spatial value and range of CO and $PM_{2.5}$ concentrations in the PBL (altitude lower than 1000 m) over three regions: Coastal region including Cotonou (6.3°N - 9.0°N), Sudano-Guinean region including Djougou (9.0°N - 11.0°N), Sudano-Sahelian region including Niamey (11.0°N - 13.5°N).

For CO concentration, on 13 June, there is no clear gradient over the three regions but rather a constant concentration of about 170 ppb. On 14 June, a gradient is noticed from the coast (200 ppb) to the Sahel (167 ppb). For both days, the model predict an opposite gradient with the highest over the Sahel. Over the coastal region, the observed CO concentration range is similar for the two flights (between 147 - 222 ppb), which is well in agreement with the modeled range (between 175 - 240 ppb). Over the Sudano-Guinean region, the observed range of variation is 161 - 182 ppb prior to the MCS (13 June), and it

increases to 153 - 233 ppb after the MCS (14 June). The model is able to capture the larger range on 14 June than on 13 June (205 - 247 ppb compared with 218 - 245 ppb). Over the Sudano-Sahelian region, the observed range of CO concentration is also larger on 14 June (146 - 200 ppb) than on 13 June (149 - 174 ppb). This behavior is not reproduced in modeled concentrations.

For the $PM_{2.5}$ concentration, a South-North gradient is expected moving closer to the Sahara. There is a clear gradient in the observed $PM_{2.5}$ concentration mean on 13 June: 8 $\mu$g.m$^{-3}$ for the coastal region, 50 $\mu$g.m$^{-3}$ for the Sudano-Guinean

region, 56 $\mu$g.m$^{-3}$ for the Sudano-Sahelian region. After the MCS, there is no clear gradient but rather the same concentration over the coastal and the Sudano-Guinean regions (39 $\mu$g.m$^{-3}$) and a gap of concentration over the Sahel (up to 92 $\mu$g.m$^{-3}$). The ranges are increased over the three region: 37 - 42 $\mu$g.m$^{-3}$ for the coastal region, 24 - 59 $\mu$g.m$^{-3}$ for the Sudano-Guinean

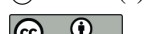



region, 50 - 139 $\mu$g.m$^{-3}$ for the Sudano-Sahelian region. The modeled ranges match the observed one for both days. The model reproduce a South-North gradient on 13 June, which is well in agreement with the observations. On 14 June, the model predict the concentration gap between the coastal and the Sudano-Guinean regions (from 43 to 82 $\mu$g.m$^{-3}$), while it was observed

between the Sudano-Guinean and the Sahelian regions.

The 13 and 14 June 2006 correspond to disturbed meteorological conditions, which may not be representative of the typical average concentrations. The model-observation comparison suggest that the MCS is not well reproduced, which could in turn induce a not realistic modeled pollution plume (for instance a biomass burning plume) over the Sudano-Sahelian region and the Sahel. Nevertheless, there is a over-estimation of the modeled CO concentration (positive bias of $\approx$ 20 ppb) for these two

days.

### 3.3.2 Monthly modeled pollution sources apportionment

In order to analyze the source apportionment, we consider that the CO mixing ratio is due to three major sources: background, anthropogenic and fires, and the PM$_{2.5}$ mass concentration comes from five major types of pollution source: anthropogenic, fires, mineral dust, biogenic and sea salt (we assume that PM$_{2.5}$ background concentration is negligible). For the whole period

and for each month of the simulation, the relative percentage of each source is presented in Table 2 and Table 3 at the three studied locations: Cotonou, Djougou and Niamey.

For the three sites, the average concentrations of surface CO increase during the whole period. The mean concentrations are very close for the three sites: 221 ppb in Cotonou, 227 ppb in Djougou, 212 ppb in Niamey. There is a clear increase of the CO from May (157 - 180 ppb) to July (267 - 280 ppb). This increase is due to the vegetation fire sources from May (3 - 10

%) to July (40 - 49 %) while the anthropogenic and background concentrations are stable during the whole period and for the three sites. It seems that the CO overestimation noticed in the previous section is linked with an overestimation of vegetation fire emissions.

Considering PM$_{2.5}$ concentrations, there is on average over the whole period a South-North gradient of concentrations (30 $\mu$g.m$^{-3}$ in Cotonou, 38 $\mu$g.m$^{-3}$ in Djougou, 54 $\mu$g.m$^{-3}$ in Niamey), consistent with the gradient of the dust contribution (15

% in Cotonou, 35 % in Djougou, 67 % in Niamey). From May to July and for the three sites, the mineral dust contribution is in constant decrease. On the other hand, the vegetation fires increases (3 to 19 $\mu$g.m$^{-3}$ in Cotonou, 1 to 17 $\mu$g.m$^{-3}$ in Djougou, 0.6 to 11 $\mu$g.m$^{-3}$ in Niamey), which could be overestimated as for CO concentration. PM$_{2.5}$ concentrations are dominated by natural sources. Nevertheless anthropogenic PM$_{2.5}$ concentrations range between 3 to 5 $\mu$g.m$^{-3}$, which is about 10 % for the whole period and for the three sites.

In Cotonou, the average concentrations of surface PM$_{2.5}$ increases during the whole period, from 23 to 37 $\mu$g.m$^{-3}$. This mainly corresponds to the arrival of vegetation emissions products, transported from Central Africa to the Guinean Gulf, with an increase from 11 to 52 % from May to July. On the contrary, the mineral dust contribution decreases during the period, from 38 to 5 %. The sea salt contribution increases from 3 to 6 ppb. During the three months, the anthropogenic and biogenic contributions remain stables at about 4 ppb and 6 ppb respectively.



In Djougou, the same behavior is observed but with some changes in the absolute values. The relative contribution of mineral dust decreases from 57 % to 14 %, while the fire contribution increases from 3 to 47 %. The anthropogenic contribution is slightly higher in June at about 5 ppb.

In Niamey, the dust contribution is important for the three months. It decreases by a factor 4, from 61 to 15 $\mu$g.m$^{-3}$, consistently with observation of PM$_{10}$ in Banizoumbou in Niger (Marticorena et al., 2010), which is probably due to the reduction of local emission linked with the increase of vegetation cover. The relative contribution of anthropogenic pollution is slightly higher in June at about 5 ppb.

For CO or PM$_{2.5}$ concentrations, the anthropogenic contribution is always important in the total budget ($\approx$ 30 % for CO and $\approx$ 10 % for PM$_{2.5}$). It is therefore important to better understand the daily variability of anthropogenic pollutants transport.

## 4    Focus on anthropogenic pollutants from Cotonou to Niamey

This section focuses on anthropogenic pollution horizontal variability and vertical structure. Only the contribution of anthropogenic sources is considered in PM$_{2.5}$ and CO concentrations, from now on referred to as anth-PM$_{2.5}$ and anth-CO.

### 4.1    Time-latitude variability at the surface

#### 4.1.1    CO and PM$_{2.5}$ concentrations

The Cotonou-Niamey meridional transect includes two specific cities extensively studied in the framework of the AMMA program: a coastal megacity (Cotonou in Benin) and a Sahelian city (Niamey in Niger). To highlight the latitudinal regional transport, modeled concentrations are presented with the same methodology as the previous section with Hovmöller diagrams, corresponding to time-latitude average of variables (data are smoothed with a 5-day moving average, *i.e.* ±2 days).

Results are presented in Figure 4 for anth-CO and anth-PM$_{2.5}$. For the two species, meteorological parameters are superimposed to the concentrations. The precipitation rate contours are defined for events with more than 10 mm/day over the Cotonou-Niamey transect. These are similar but not equivalent patterns to those presented in section 3.1 averaged over the entire West Africa.

The Inter-Tropical Discontinuity (ITD) is the limit between the northward monsoon wind and the southward Harmattan wind (Flamant et al., 2007; Karam et al., 2009). The ITD could be defined as the isocontour of relative humidity (RH) equal to 20 %. We can notice that the location of the ITD marks a sharp gradient in surface concentrations, with a decrease from 100 to 20 ppb for CO, and from 5 $\mu$g.m$^{-3}$ to 2 $\mu$g.m$^{-3}$ for PM$_{2.5}$.

For anth-CO surface concentrations, high concentrations are continuously noticed during the whole period at the coast where Cotonou is located, about 100 ppb around $\phi$ = 6.3°N. Over the Guinean gulf, the concentration is low between 20 and 50 ppb. The second area with high anth-CO values corresponds to the Sudano-Sahelian region, where concentration vary between 60 and 100 ppb around $\phi$ = 12°N. Whereas high concentrations of anth-CO at the coast is clearly related to local



emissions, the high concentrations over the Sahel should be either due to transport, or local emissions. Over the studied

domain, anth-CO surface concentrations evolves between 20 and 120 ppb. There are in May and in June some high modeled surface concentrations when rain occurs such as around 11 June. In July, the variability is mostly consistent with precipitation rates after the onset, which suggest that surface versus vertical distribution has changed by the convection associated with large scale precipitation. The precipitation variability can thus explain only a part of the CO variability. It is necessary to also investigate the large-scale wind speed and direction.

The same behavior is observed for the surface concentrations of $PM_{2.5}$. The week to week variability is more important. This increase is probably due to the longer CO lifetime compared with that of PM (being less chemically active and without settling), the CO concentrations are more homogeneously mixed in a large latitudinal area from the coast to more than $16°$ up to the North. The temporal variability of surface $PM_{2.5}$ exhibits a frequency close to 2-week: during the whole modeled period, five higher concentration periods are observed from the coast to $\phi \approx 16\ °N$. In addition in these latitudinal plumes,

local minima are modeled, for instance the 20 June at $\phi \approx 13\ °N$. At $\phi = 12°N$, there is an area of high concentration, which is present over the whole period. This may be related to vertical transport and will be quantified in the next sections. The results showing similarities for both two pollutants in terms of time-latitude variability, the next sections will refine the analyzes only for $PM_{2.5}$.

### 4.1.2   Synoptic wind and pollution

This section aims at analyzing the anth-$PM_{2.5}$ concentration temporal frequency through the surface wind speed and direction. The previous section has shown that low pollution levels are not always associated with precipitations. Results are presented in the same way using Hovmöller diagrams on Figure 5. For each figure, colored isocontours are superimposed corresponding to surface anth-$PM_{2.5}$ of 4 and 5 $\mu g.m^{-3}$.

For the surface wind speed, the lowest values are modeled along the coast during the whole period. The wind speed variability

is weak from the coast to the Sahel. Periods of low wind speed are coincident with the highest values of surface $PM_{2.5}$. At the end of the period, the meteorological situation changes and the wind speed increases suddenly over the ocean showing the cold tongue arrival as previously described by Meynadier et al. (2016), when there are precipitations inland and low anthropogenic pollution levels.

Regarding the precipitations occurrences discussed in the previous section, the high surface anth-$PM_{2.5}$ concentrations

modeled around $\phi = 12°N$ are due to a combination of low wind speed and low precipitation rates. These meteorological conditions are representative of stagnation, which accumulate pollutants in the lowest layers of the troposphere.

For the surface wind direction, the main wind direction near the coast is the South-West quarter during the whole period. There is no obvious link between wind direction changes and $PM_{2.5}$ highest values over the Cotonou-Niamey meridional transect.

The large scale variability of meteorological variables (precipitation and wind speed) controls the period of high anthropogenic pollution from the coast to the Sahel. However, it does not explain if the high concentration over the Sahel are linked with local emissions or/and pollutants transport from the coast.



## 4.2 Monthly mean vertical structure

### 4.2.1 Anthropogenic pollution

We now focus on the vertical structure of the lower troposphere from the surface to 4 km altitude in order to understand what is responsible for the high anth-$PM_{2.5}$ over the Sahel. Monthly averages of anth-$PM_{2.5}$ concentration are analyzed together with wind circulations (monthly averages correspond to consistent meteorological periods, *c.f.* Section 3.1). In Figure 6, the modeled concentrations are spatially averaged over the Cotonou-Niamey meridional transect. Three isocontours (3, 4 and 5 $\mu g.m^{-3}$) are used to follow the anthropogenic pollution patterns. Results are presented at 00 UTC when the NLLJ is established (on the contrary, during the day, the pollution is mixed in the PBL by dry convection).

For the three months, the meridional wind is lower at the surface than in the boundary layer from the coast to $\approx 9°N$, highlighting the well established NLLJ. Above the northward monsoon flux, there is the SAL associated with southward winds. The highest southward wind speed in the core of the SAL between 11 and 16°N in latitude and 2 to 4.5 km in altitude is the African Easterly Jet (AEJ).

Regarding the wind, two atmospheric cells are noticed on Figure 6 during the three months. There is a large cell going northward at the surface within the monsoon flow, and going backward toward the South with the SAL (or AEJ), located at $\approx$ 2 km altitude, and between the coast and $\approx 16°N$. There is also a small cell turning in the same direction (clockwise) at $\approx$ 2 km altitude, with the downdraft at 6°N and the updraft at 7°N. In May and June, the small cell is included in the large cell, while in July, they are disconnected. These two cells seem to interact with the anthropogenic pollution because the anth-$PM_{2.5}$ isocontours shape appear to be driven by the wind patterns.

Regarding anth-$PM_{2.5}$ concentration, these two atmospheric cells seem to interact with the anthropogenic pollution because the anth-$PM_{2.5}$ isocontours shape appear to be driven by the wind patterns. The large atmospheric cell induces a recirculation of the modeled anthropogenic plume, ranging from $\phi = 6$ to 18°N in latitude and 0.5 to 3 km in altitude (anth-$PM_{2.5}$ isocontour of 3 $\mu g.m^{-3}$). We can notice that the recirculation center is located at $\phi \approx 14$ °N during the three months studied. In May, an important part of the pollution from the coast is transported in altitude within the NLLJ above the PBL (displayed by the anth-$PM_{2.5}$ isocontour of 4 $\mu g.m^{-3}$). In June, there is high concentration in altitude over the Sahel (displayed by the anth-$PM_{2.5}$ isocontour of 5 $\mu g.m^{-3}$), which suggests that the atmospheric cell concentrates pollutants. Note that in July, the lowest latitude of the anthropogenic plume is moved northward, starting at $\approx 10°N$, and less connected with the coast.

Air masses transport anthropogenic pollutants from the coast to the Sahel. High surface concentrations of anth-$PM_{2.5}$ are modeled at the latitude of the coastal urbanized areas ($\phi = 6.3°N$), leading to a plume to the North within the NLLJ and concentrating pollutants over the Sahel.

### 4.2.2 Coastal versus Sahelian pollutants meridional transport

A tracer experiment has been set-up to analyze if the main contributors to the Sahelian maximum are emitted locally or remotely at the coast. Gaseous tracers are released at the two major cities of the meridional transect (without any sink): Cotonou (Benin) and Niamey (Niger). The tracers are constantly released from the 1 to 30 June. The emission altitude occurs in the PBL (0 -



500 m) from the 1 June to 30 June. Results are presented in Figure 7 averaged during the ten last days at 00 UTC considering only coastal emissions or only Sahelian emissions in order to observe where air recirculation may concentrate the pollutants.

Tracers emitted at the coast indicate that there is a contribution of coastal pollutants over the Sahel. The ratio of Sahelian tracers concentration and coastal tracers concentration in the Sahel is between 10 % and 1 %; *i.e.* with coastal emission 10 to 100 times higher, the anthropogenic coastal pollution could be equivalent to the anthropogenic Sahelian pollution over the Sahel. In the HTAP anthropogenic inventories (presented in Figure 1), the anth-$PM_{2.5}$ (respectively anth-CO) is at Niamey $\approx$ 103 (735) $\mathrm{kg.km}^{-2}.\mathrm{day}^{-1}$ and at Cotonou $\approx$ 438 (7707) $\mathrm{kg.km}^{-2}.\mathrm{day}^{-1}$. Therefore, an important part of the pollution over the Sahel has been emitted at the coast, which contributes to a maximum of anthropogenic pollution in June. In conclusion, the high concentration in altitude over the Sahel are due to atmospheric cells, which concentrate pollutants emitted locally and remotely at the coast.

## 5   Impact of coastal dynamics on anthropogenic pollution

In order to better characterize the coastal pollution, anthropogenic $PM_{2.5}$ for the period of 8 to 15 June 2006 are now described at hourly temporal resolution. This week includes a large variability of low to high surface concentrations.

### 5.1   Surface hourly pollution variability

The surface hourly anthropogenic $PM_{2.5}$ concentrations are shown on Figure 8 over the Cotonou meridional transect. The highest temporal resolution shows the same variability as described in Figure 4 with the beginning of the week associated with lower concentrations.

For most days, the highest surface concentrations are modeled between 18 UTC and 00 UTC from the coast to 8°N. This coincides with the lowest boundary layer height, which concentrates urban emissions (i.e waste burning and traffic) in a thin layer (not shown). It seems that urban plumes start from the coastline, corresponding to the city, at the sunset (18 UTC) and derives to the North in the next few hours.

At the coast, anth-$PM_{2.5}$ concentration decreases at night (between 00 UTC and 06 UTC) and, in the morning (around 06 UTC), concentration increases again at the surface when the convection and NLLJ are weak, which follows traffic emissions.

There is a transition from low to high level of anthropogenic pollution from the 8 to 12 June. It is interesting to note on the 11-12 June night that high concentrations are modeled during the day when there is precipitation inland. This precipitation event reflects a change in the wind patterns, which induces a transport of pollutants over the sea (with surface concentrations up to 3 $\mu\mathrm{g.m}^3$).

### 5.2   Contribution of other cities at Cotonou

At the coast, the wind direction comes from the sector S-SW but there are diurnal variations, which could affect the air pollution along the coastline. In order to distinguish pollutants transport from the different coastal megacities, a tracer experiment has been set-up. The tracers are constantly released in the lowest model layers ranging from surface to 500 m above sea level.





This altitude corresponds to emission below or within the NLLJ. Three point sources have been defined: Accra in Ghana (5.6°N,0.2°W), Lome in Togo (6.2°N,1.2°E) and Cotonou in Benin (6.4°N,2.4°E). The results are presented over the Cotonou meridional transect (Figure 9). We aim at evaluating the impact of the Cotonou local emissions versus emissions from distant

areas transported toward Cotonou. The emission (in arbitrary unit) has the same magnitude in each city without any daily variation.

As expected, the surface concentrations of Cotonou due to local emission only are the highest at the coast. The diurnal cycle of pollutants transport appears clearly with the highest concentrations exported at the beginning of the night (18 UTC), when the boundary layer height decreases quickly and when the establishment of the NLLJ occurs. Up to 9°N, high tracers

concentrations are transported from Cotonou far from the point source. The Cotonou plume always transports tracers up to 7°N; for some days, these plumes may reach the latitude of 9°N during the night between 18 UTC and 06 UTC.

At Cotonou, the modeled tracers concentration released from Lome or Accra are logically lower because the source points are not in the Cotonou meridional transect studied. The same kind of northward transport is observed but pollutants transport from Accra and Lome reach Cotonou in the morning between 06 UTC and 12 UTC for the Lome plume, and in the afternoon between

12 UTC and 18 UTC for the Accra plume. This result is consistent with a transport speed between 10 and 20 km.hour$^{-1}$ (the Lome-Cotonou distance is about 150 km, and Accra-Cotonou about 300 km). Moreover, there is probably a diurnal variation of the wind direction, coming from the sector S-SW during the day and SW-W at night because of the land/sea breeze, which carry more efficiently pollutants emitted during the night toward Cotonou.

In arbitrary unit and with the same emission at the three cities, the concentrations modeled at latitude between 7°N and 8°N

are similar for the Lome and Cotonou plumes, and in a lower extent for the Accra plume. This means that the urbanized areas West from Cotonou (Lome or Accra) contribute as much as Cotonou to the anthropogenic pollution at those locations (when considering the same emission).

The comparison between the three plumes shows a specific behavior during 10-12 June as the Cotonou pollution is not exported to the North:

– On 10 June, the Lome plume is clearly exported over the sea and very high concentrations are noticed over Cotonou. The same behavior is observed for the Accra plume with lower concentration because this location is further from Cotonou. Indeed, the Lome and Accra plumes reach Cotonou after being transported over the sea, which suggests that pollutants at the three cities have been transported Eastward, leading to plumes overlaying at the Cotonou location.

  – On 11 June, there are still high concentrations over Cotonou due to the Accra plume probably driven by the same meteorological conditions, but the Lome plume does not have a specific behavior. This suggests a perturbation affecting especially Accra.

  – The 12 June corresponds to the most important transport of Cotonou emission to the North. At the same time, this is the

only day when Lome and Accra plumes do not reach Cotonou, as they are shifted to the North at 7°N when crossing the Cotonou meridional transect.



All these results suggest a fast change of the meteorological situation, leading to air pollution. In the next section, the specificity of the vertical wind structure during this period will be studied in detail.

## 6 Disturbed atmospheric dynamics and pollution transport

In this section the analysis is refined to two periods of two days on 8-9 and 11-12 June 2006. The first corresponds to non-perturbated monsoon flow situation leading to low anthropogenic pollution, the second corresponds to a disturbed situation leading to high anthropogenic pollution.

### 6.1 Evolution of the vertical structure

In order to focus the day/night transport from the coast to the North (described in section 5.1) and the changes in dynamical
regimes during this period (described in section 5.2), results are presented as vertical slices in Figure 10, averaged along the Cotonou meridional transect. The tracers concentrations, emitted separately at Lome and Cotonou, are presented as isocontours of threshold values: the emissions being arbitrary, the modeled concentrations are also arbitrary. But the same threshold is used for the two emissions locations, thus concentration magnitudes are comparable. The wind vectors (meridional/vertical components) are superimposed on the figures to highlight the vertical cells. The meridional wind is also presented as color
shading for the NLLJ intensity.

The first period, 8 June at 23 UTC and 9 June at 11 UTC, corresponds to a classical monsoon case, often observed and described in the literature (Abdou et al., 2010; Lothon et al., 2008). At night, surface pollutants are concentrated in a shallow layer (less than 200m), corresponding to nocturnal surface layer and to the lowest part of the NLLJ (represented by the 'dark blue' shaded area in Figure 10). During the day, the convection induces more mixing over land than over sea, where the
boundary layer reaches 1500 m at 11 UTC. The Lome plume does not reach the coast, but it crosses the Cotonou meridional transect more to the North > 6.5°N.

On 8-9 June, an updraft-downdraft convective cell is clearly observed during the day and at night, with ascendent wind at 7°N and subsident wind at 6.3°N (the Cotonou site latitude). This circulation has already been observed for the whole studied period in section 4.2.1. This is not a modeled land-sea breeze because it turns in the same direction day and night. Land-sea
breezes have not been explicitly modeled because of the too coarse resolution (about 20 km).

The second period, 11 June at 23 UTC and 12 June at 11 UTC, corresponds to a disturbed case compared to what is usually observed in this region and for this month (for instance 8-9 June). Indeed, for the 11 June at 23 UTC, the NLLJ is not present near the coast and the wind is weak from the coast to 8°N. The modeled nocturnal PBL height is very low (less than 50m). The Lome plume is not present over Cotonou. On 12 June at 11 UTC, air subsidence is modeled from 7°N to 10°N. The isocontours
of concentrations due to emissions in Lome and Cotonou are at the same latitude, corresponding to an iso-latitudinal transport, along the coast. Compared to the 8-9 June, there is no coastal cell located over the emission region. But a larger cell is modeled, with high meridional wind speed (up to 4 m.s$^{-1}$). The subsidence, located at $\phi > 7$°N, imports upper air masses from the free troposphere and blocks the northward transport of the coastal pollutants.





## 6.2 Specificity of 11-12 June 2006

In this last section, we focus on the 11-12 June to understand which meteorological conditions have led to an important modeled anthropogenic PM$_{2.5}$ event at Cotonou. Schwendike et al. (2010) have shown that a large MCS has occurred over Ghana due to convective instabilities at the border of Togo, Ghana and Burkina Faso. Some spots of convection ('Pop corn' convection) over a large region including Cotonou has been identified on 11 June. An isolated convective cell lasting a few hours coming from South-East and going North-West have crossed the coastline over Cotonou at around 18 UTC (Figure **??**), which is well in agreement with the modeled location (Figure 12). When precipitation is inland between 19 UTC and 23 UTC (Figure 12 - top), the wind speed is null over the coast because the monsoon flux is blocked. During these specific meteorological conditions, the high anth-PM$_{2.5}$ surface concentrations are thus due to an accumulation of pollution during a few hours (from 19 to 23 UTC).

The same tracer experiment as described in the previous sections (described in section 5.2) is used to test our hypothesis about the accumulation of pollutants and to distinguished plumes of the different cities. Gaseous tracers are released with the same emission at four cities: Accra (Ghana), Lome (Togo), Cotonou (Benin) and Lagos in Nigeria has been added (6.5°N,3.4°E) .

We can notice (on Figure 12 -bottom) that the pollution emitted at the different cities West from Cotonou is mixed at $\phi$ = 7°N on 11 June at 19 UTC. Six hours later, only the Cotonou plumes is responsible for the high anth-PM$_{2.5}$, which confirms pollutants accumulation because it has been blocked by the down-draft of the precipitation system.

This result demonstrates that during the monsoon period, specific meteorological conditions could lead to high pollution in the Guinean coast megacities, although most of the time pollution emitted along the coastline are quickly transported to the North.

## 7 Conclusions

The West African pollution has been studied using both models and observations during May, June and July 2006. This corresponds to the beginning of the WAM and includes the AMMA campaign observational period. The focus is on urbanized areas, located along the Guinean Gulf and known as large gas and aerosol emitters. In addition to these anthropogenic emissions, the coast is often under the influence of long-range transport of mineral dust and vegetation fires emissions. The analyses are performed for CO and PM$_{2.5}$ over a large domain to include all sources: Central Africa for biomass burning, Sahel and Sahara for mineral dust and a large part of the Guinea Gulf for sea salt.

The first analysis was devoted to estimate the relative contribution of each source during the three months in Cotonou (Benin), Djougou (Benin) and Niamey (Niger). It was shown that the surface concentrations of PM$_{2.5}$ constantly increase during the period. The mineral dust dust relative contribution remains low close to the coast, showing that in monthly average, pollution during this period is not dominated by mineral dust transport events. On the other hand, the vegetation fires increase from May to July. The anthropogenic part is stable during the whole period for the three studied sites at $\approx$ 50 % for CO and $\approx$ 15 % for PM$_{2.5}$.

The second part of the study was focused on the anthropogenic contribution of CO and PM$_{2.5}$ along a Cotonou-Niamey meridional transect. A transport of these pollutants from the coast to the north has been demonstrated up to the Sahel (13°N).





The Northward limit of the transport corresponds to the Inter Tropical Discontinuity. It was also shown that there are alternating periods of high/low concentration from Cotonou to Niamey with a weekly frequency. To understand this variability, meteorological variables have been investigated. The highest surface pollutants concentrations occurred when there is no precipitation and low wind speed.

In order to better understand the meridional transport and the occurrence of high pollutants concentrations over the Sahel ($\approx$ 13°N), monthly averages of vertical wind structure were analyzed. From May to June, a large atmospheric cell going from the coast to the Sahel remains present and it has been identified as responsible for the accumulation over the Sahel of pollutants emitted locally and remotely at the coast.

    A focus has been done on coastal dynamics and pollution transport during a restricted period, from 8 to 15 June 2006,
including high and low levels of anthropogenic pollution. To isolate the coastal dynamics impacts on several cities plumes from the coastline, a tracer experiment was designed with emissions at Accra (Ghana), Lome (Togo) and Cotonou. The tracer concentrations confirm that, in Cotonou, the modeled concentrations are both due to local and remote emissions. A meridional transport of the anthropogenic pollution from the coast to the North has been highlighted at night linked with the Nocturnal Low Level Jet close to the coast.

Finally, two contrasted anthropogenic pollution situations were detailed. The first situation (8-9 June) corresponds to low anthropogenic pollution during a 'typical' case of monsoon dynamics, while the second situation (11-12 June) corresponds to a disturbed meteorological situation due to a MCS. During the 11-12 June, it was shown that air subsidence is modeled at latitude 7°N, which imports clean upper air masses from the free troposphere, limiting the northward transport of the coastal pollutants. The main results of this article will be compared to 2016 DACCIWA campaign observations in order to confirm and
refine your conclusions.

*Acknowledgements.* The research leading to these results has received funding from the European Union 7th Framework Programme (FP7/2007-2013) under Grant Agreement no. 603502 (EU project DACCIWA: Dynamics-aerosol-chemistry-cloud interactions in West Africa). This work has been supported by the African Monsoon Multidisciplinary Analysis (AMMA) project. Based on a French initiative, AMMA was built by an international scientific group and is currently funded by a large number of agencies, especially from France, UK, USA and various African countries. The authors wish to thank the SAFIRE (Service des Avions Francais Instruments pour la Recherche en Environnement) for preparing and delivering the research aircrafts (ATR-42). We thank the Philippe Goloub and Didier Tanre for theirs
effort in establishing and maintaining AERONET sites in Djougou (Benin) and in Banizoumbou (Niger).



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



## List of Figures









**Figure 1.** *Anthropogenic carbon monoxide (top) and primary particulate matter (bottom) surface emission fluxes in $kg.km^{-2}.day^{-1}$ averaged over the three months period. The gray dots are the major cities and the three green dots are the locations studied from the South to the North: Cotonou (Benin), Djougou (Benin), Niamey (Niger). The blue box represents the Cotonou (Benin) - Niamey (Niger) meridional transect studied (longitude $\lambda = 2°$ East to $3°$ East, latitude $\phi = 1°$ North to $19°$ North).*




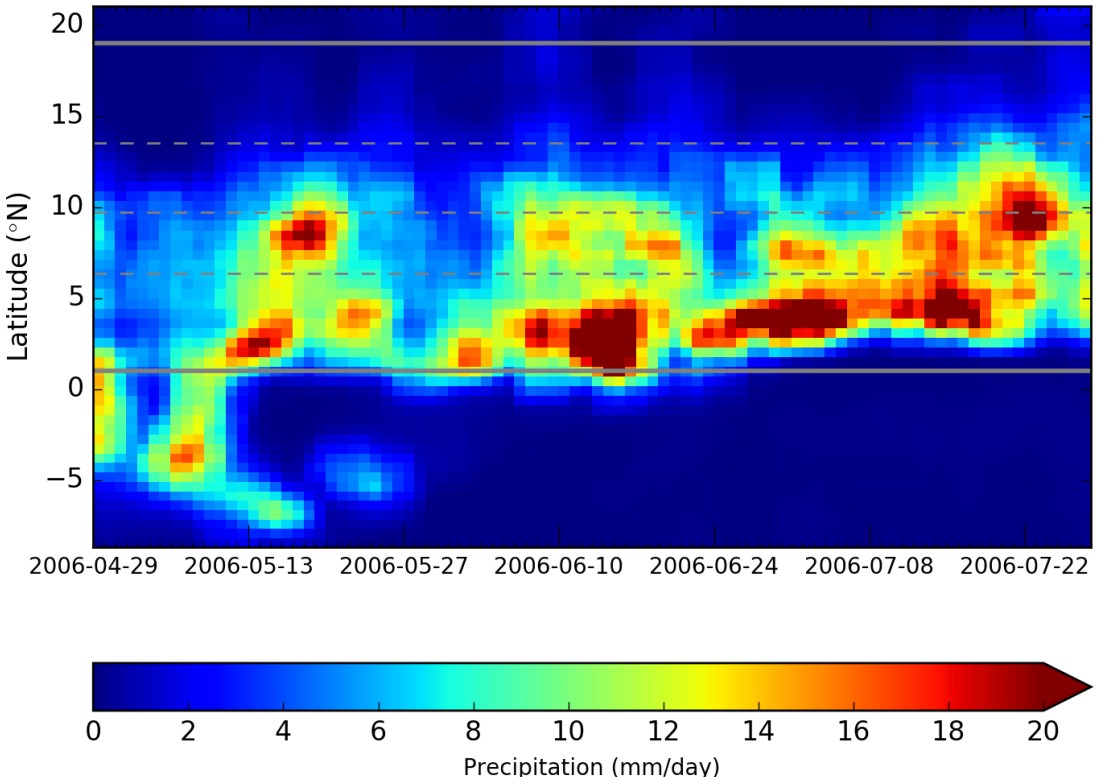

**Figure 2.** *Time-latitude average (Hovmöller) of precipitation (mm.day$^{-1}$). Precipitation is averaged between 8.5°W and 8.5°E in longitude. Day-to-day variability is eliminated by applying a moving average of ±2 days. Due to the longitudinal average, the coastline is between 5°N and 6°N. The 2 gray lines show the latitudinal extend of the regional domain. The 3 dash gray lines present the latitudes of the 3 locations studied (Cotonou, Djougou and Niamey).*





**Figure 3.** *Observed daily averages of AERONET level 2 AOD and Angström exponent (black dots) at Djougou (Benin) and Banizoumbou compared to the modeled time series with a splitting to extract the relative contribution between anthropogenic, biogenic and mineral dust (all three in blue) and biomass burning emissions (in red).*





**Figure 4.** *Time-latitude average (Hovmöller diagram) of: a) surface CO concentration (ppb), and b) $PM_{2.5}$ ($\mu g.m^{-3}$) due to anthropogenic emissions along a meridional transect from 2°N to 19°N and averaged from 2°E to 3°E including Cotonou (Benin) and Niamey (Niger). Day-to-day variability is smoothed by applying a moving average of ±2 days. White contours are precipitation = 10 mm.day$^{-1}$. Specific isocontours are highlighted = 4 $\mu g.m^{-3}$ (orange) and = 5 $\mu g.m^{-3}$ (pink). The black line is the ITD defined as RH isocontour = 20 %.*





**Figure 5.** *Time-latitude average (Hovmöller diagram) of surface wind: a) speed and b) direction, along a meridional transect from $2°N$ to $19°N$ and averaged from $2°E$ to $3°E$ including Cotonou (Benin) and Niamey (Niger). Wind directions are presented with steps of $45°$. Day-to-day variability is smoothed by applying a moving average of $\pm2$ days. Orange and pink contours are surface $PM_{2.5}$ concentrations of 4 and 5 $\mu g.m^{-3}$.*



**Figure 6.** *Vertical cross-section of the meridional wind (shading in m.s$^{-1}$) mean over: a) May, b) June and c) July, at 00 UTC along a meridional transect from 2°N to 19°N and averaged from 2°E to 3°E including Cotonou (Benin) and Niamey (Niger). Light orange, orange and pink shading represents anthropogenic PM$_{2.5}$ concentration =29.0 μg.m$^{-3}$, = 4.0 μg.m$^{-3}$ = 5.0 μg.m$^{-3}$. Vectors are vertical and meridional wind (with an aspect ratio of 500). The green line is the PBL height (m). The grey vertical dash line is the latitude of the coast.*



**Figure 7.** *Vertical cross-section of the meridional wind (shading in m.s$^{-1}$) along a meridional transect from 2°N to 19°N and averaged from 2°E to 3°E including Cotonou (Benin) and Niamey (Niger), averaged over 20 to 30 June at 00 UTC. Isocontours represent gaseous tracers concentration continuously emitted (in arbitrary unit) from the 1 to 30 June at: a) Niamey (Niger) and b) Cotonou (Benin). Brown, yellow and white shading represent tracers concentration = 1 a.u., = 10 % a.u. and = 1 % a.u.; The green line is the PBL height (m). The white vertical dash line is the latitude of the coast.*



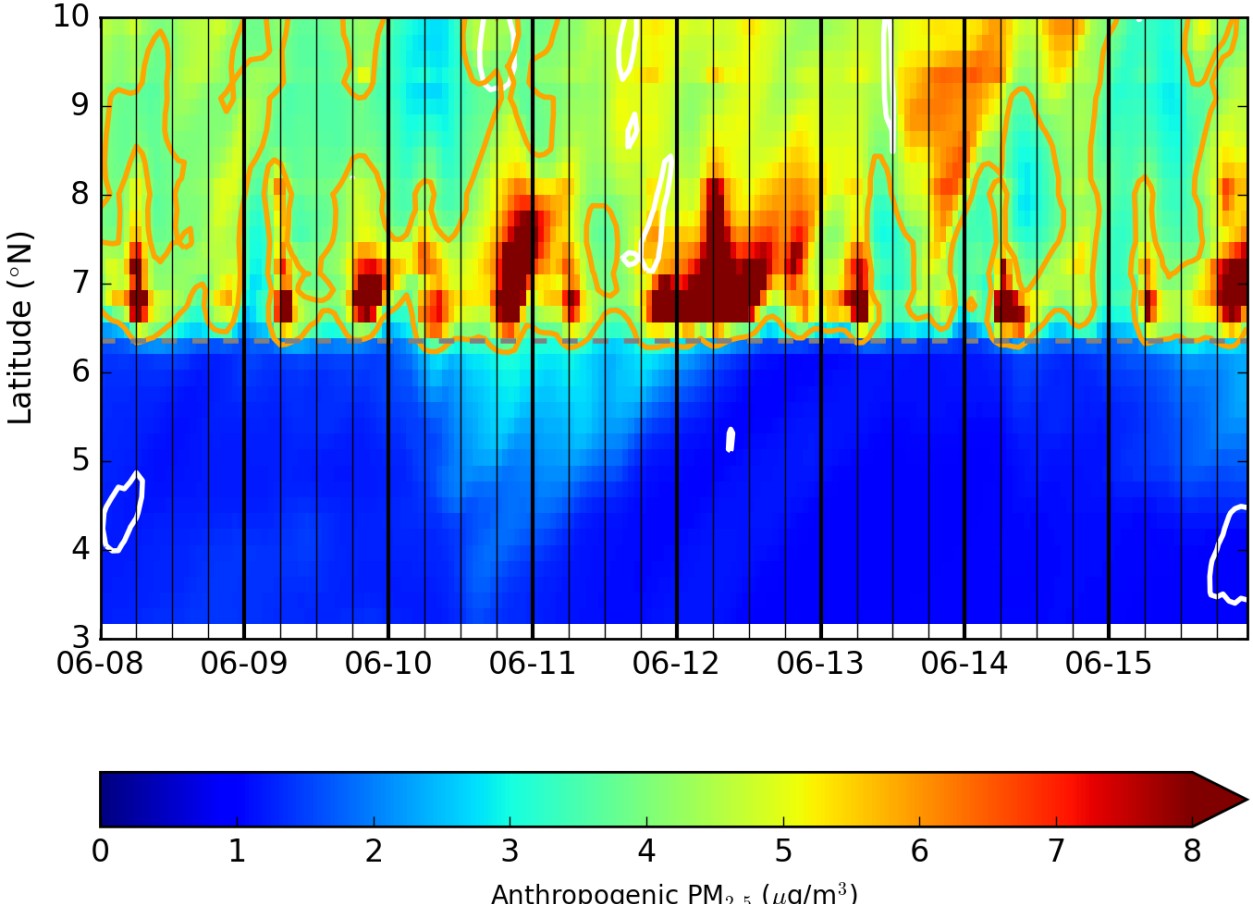

**Figure 8.** *Time-latitude average (Hovmöller) of surface anthropogenic PM$_{2.5}$ concentration ($\mu$g.m$^{-3}$) averaged along a meridional transect between 2°East and 3°East from 1 to 17 June 2016. Black vertical bars delimit the periods of the day (00 UTC; 06 UTC; 12 UTC; 18 UTC). White isocontours present precipitation rate = 3.0 mm.hour$^{-1}$). Orange isocontour represents the surface anthropogenic PM$_{2.5}$ concentration = 4 $\mu$g.m$^{-3}$.*





**Figure 9.** *Time-latitude average (Hovmöller) of gaseous tracers concentration (a.u.) averaged along a meridional transect between 2° E and 3° E centered on Cotonou (Benin) from 8 to 15 June 2006. Emissions are set up with a constant emission between 0 m and 500 m altitude at: a) Cotonou (Benin), b) Lome (Togo) and c) Accra (Ghana). Black vertical bars delimit the periods of the day (00 UTC; 06 UTC; 12 UTC; 18 UTC).*






**Figure 10.** *Vertical cross-section of the meridional wind (shading in $m.s^{-1}$) along a meridional transect from $5°N$ to $10°N$ and averaged from $2°E$ to $3°E$ including Cotonou (Benin). The two orange isocontours are tracer concentrations released in Cotonou and in Lome, respectively bold and dashed, with same threshold values (in arbitrary unit). Vectors are vertical and meridional wind (with an aspect ratio of 500). The green line is the PBL height (m). The white vertical dash line is the latitude of the coast.*



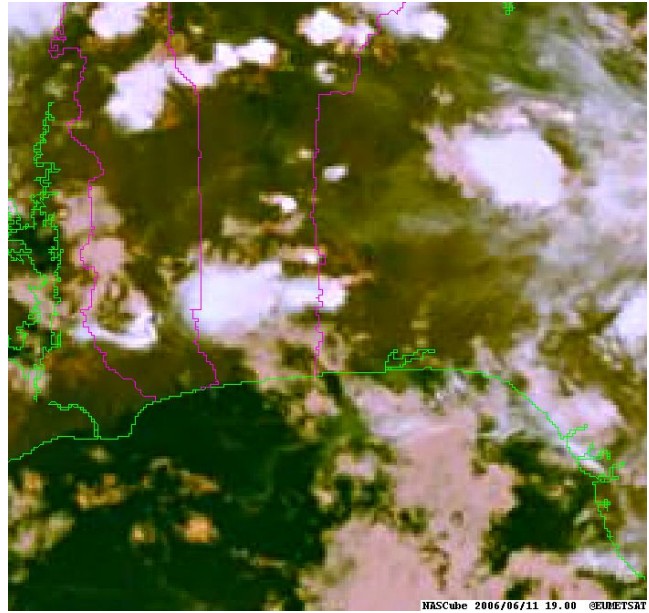

**Figure 11.** *EUMETSAT visible image of the Cotonou area of the 11 June 2006 at 19 UTC (from NAScube (http://nascube.univ-lille1.fr)*



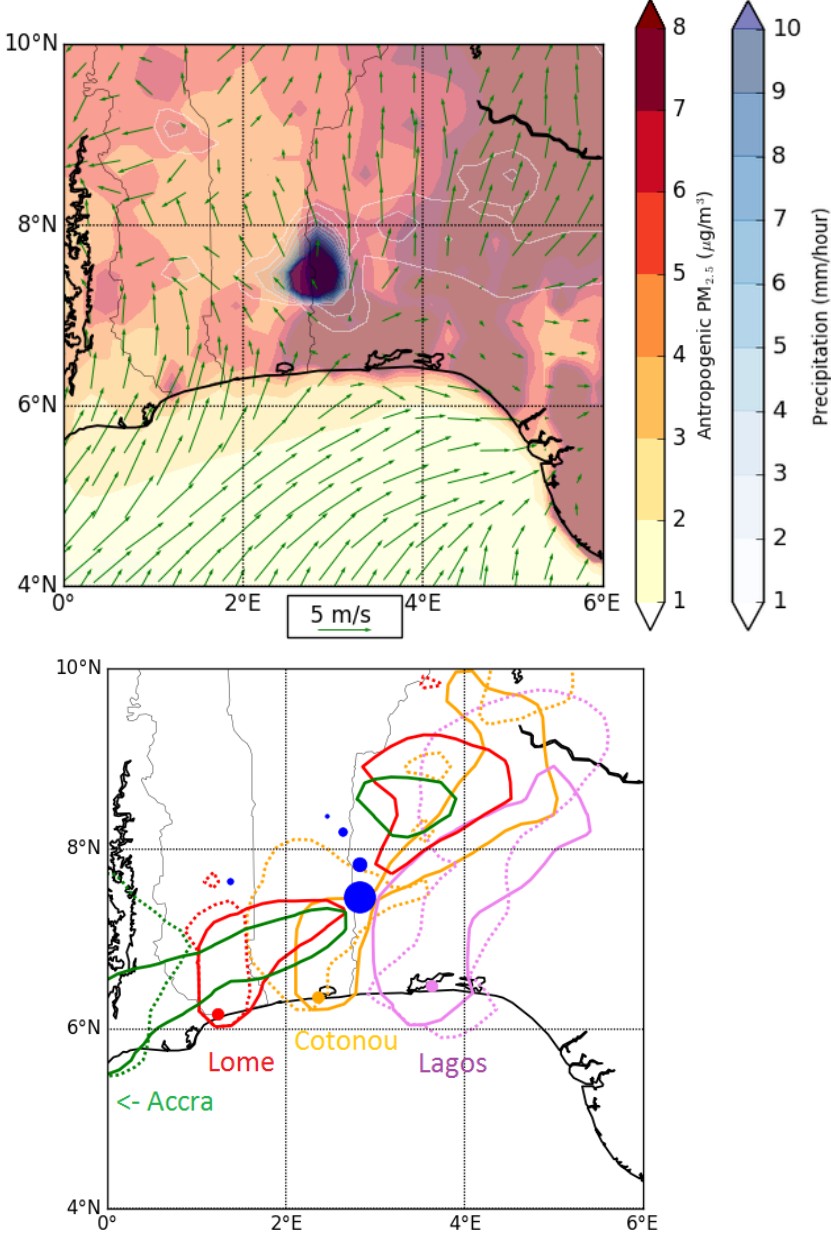

**Figure 12.** *(top) Map of Cotonou area for the 11 June 2006 at 19UTC with wind vectors at 10 m (green arrows), precipitation (blue shading), anthropogenic PM$_{2.5}$ concentration (red shading); (bottom) Isocontours of tracers concentration on 11 June at 19 UTC (solid line) and on 12 June at 01 UTC (dashed line), released in Accra (Ghana) in green, Lome (Togo) in red, Cotonou (Benin) in orange, Lagos (Nigeria) in violet. Blue dots show precipitation location each hour between 11 June at 19 UTC and on 12 June at 01 UTC (the size of blue dots depends on precipitation amount).*





**List of Tables**



| Pollutants obs/mod | Coastal region | | | Sudano-Guinean region | | | Sudano-Sahelian region | | |
|---|---|---|---|---|---|---|---|---|---|
| **Aircraft observations of 13 June 2006 from 10 UTC to 13 UTC** | | | | | | | | | |
| | Mean | Min | Max | Mean | Min | Max | Mean | Min | Max |
| CO CHIMERE (ppb) | 207.24 | 174.90 | 233.03 | 231.52 | 217.50 | 244.54 | 243.88 | 212.25 | 275.76 |
| CO Aircraft (ppb) | 172.78 | 146.78 | 209.21 | 172.51 | 161.43 | 182.14 | 159.26 | 148.70 | 174.24 |
| PM$_{2.5}$ CHIMERE ($\mu$g.m$^{-3}$) | 23.95 | 19.26 | 26.01 | 28.96 | 25.46 | 33.87 | 55.15 | 35.18 | 79.93 |
| PM$_{2.5}$ Aircraft ($\mu$g.m$^{-3}$) | 15.67 | 7.57 | 33.02 | 49.72 | 22.33 | 77.88 | 55.81 | 34.22 | 72.20 |
| **Aircraft observations of 14 June 2006 from 13 UTC to 16 UTC** | | | | | | | | | |
| | Mean | Min | Max | Mean | Min | Max | Mean | Min | Max |
| CO CHIMERE (ppb) | 212.71 | 190.23 | 239.83 | 229.15 | 205.49 | 246.75 | 244.74 | 232.68 | 257.97 |
| CO Aircraft (ppb) | 200.21 | 185.49 | 222.34 | 181.51 | 153.11 | 233.35 | 168.66 | 146.35 | 200.66 |
| PM$_{2.5}$ CHIMERE ($\mu$g.m$^{-3}$) | 42.79 | 28.23 | 64.33 | 82.25 | 64.43 | 92.93 | 84.01 | 79.65 | 91.46 |
| PM$_{2.5}$ Aircraft ($\mu$g.m$^{-3}$) | 39.40 | 37.26 | 42.37 | 39.36 | 23.60 | 59.12 | 92.08 | 50.11 | 138.86 |

**Table 1.** *Range of 2-minute average modeled and observed concentrations of CO (ppb) and PM$_{2.5}$ ($\mu$g.m$^{-3}$) in the PBL (altitude lower than 1000 m) over three regions: Coastal region (6.3°N - 9.0°N), Sudano-Guinean region (9.0°N - 11.0°N), Sudano-Sahelian region (11.0°N - 13.5°N).*




| CO | May-July | | May | | June | | July | |
|---|---|---|---|---|---|---|---|---|
| *Cotonou (Benin)* | | | | | | | | |
| **Average (ppb)** | **221.01** | | **157.25** | | **239.11** | | **267.26** | |
| Background (ppb and %) | 73.30 | 33.17 | 73.58 | 46.79 | 72.82 | 30.45 | 73.48 | 27.50 |
| Anthropogenic (ppb and %) | 64.75 | 29.30 | 67.83 | 43.14 | 64.65 | 27.04 | 61.77 | 23.11 |
| Fire (ppb and %) | 82.96 | 37.54 | 15.84 | 10.07 | 101.64 | 42.51 | 132.00 | 49.39 |
| *Djougou (Benin)* | | | | | | | | |
| **Average (ppb)** | **226.78** | | **180.28** | | **240.65** | | **259.85** | |
| Background (ppb and %) | 75.27 | 33.19 | 77.38 | 42.92 | 75.23 | 31.26 | 73.21 | 28.17 |
| Anthropogenic (ppb and %) | 80.77 | 35.62 | 93.52 | 51.88 | 89.01 | 36.99 | 60.04 | 23.11 |
| Fire (ppb and %) | 70.74 | 31.19 | 9.38 | 5.21 | 76.41 | 31.75 | 126.60 | 48.72 |
| *Niamey (Niger)* | | | | | | | | |
| **Average (ppb)** | **212.44** | | **171.24** | | **229.26** | | **237.34** | |
| Background (ppb and %) | 78.95 | 37.16 | 82.72 | 48.30 | 78.69 | 34.32 | 75.42 | 31.78 |
| Anthropogenic (ppb and %) | 82.54 | 38.85 | 83.92 | 49.01 | 98.24 | 42.85 | 65.96 | 27.79 |
| Fire (ppb and %) | 50.95 | 23.98 | 4.60 | 2.69 | 52.33 | 22.82 | 95.97 | 40.43 |

**Table 2.** *CO (ppb) average and relative contributions (%) of each type of pollution source (background, anthropogenic, fire) at Cotonou (Benin), Djougou (Benin) and Niamey (Niger). The time averaged periods correspond to each month of May, June, July and to the whole period (from May to July).*




| PM$_{2.5}$ | **May-July** | | **May** | | **June** | | **July** | |
|---|---|---|---|---|---|---|---|---|
| *Cotonou (Benin)* | | | | | | | | |
| **Average ($\mu$g.m$^{-3}$)** | **29.62** | | **23.31** | | **28.89** | | **36.64** | |
| Anthropogenic ($\mu$g.m$^{-3}$ and %) | 3.45 | 11.64 | 3.44 | 14.75 | 3.44 | 11.90 | 3.47 | 9.46 |
| Fire ($\mu$g.m$^{-3}$ and %) | 12.02 | 40.58 | 2.54 | 10.89 | 14.38 | 49.79 | 19.21 | 52.44 |
| Dust ($\mu$g.m$^{-3}$ and %) | 4.35 | 14.68 | 8.89 | 38.15 | 2.34 | 8.10 | 1.74 | 4.76 |
| Biogenic ($\mu$g.m$^{-3}$ and %) | 6.04 | 20.38 | 5.84 | 25.04 | 5.68 | 19.65 | 6.59 | 17.98 |
| Salt ($\mu$g.m$^{-3}$ and %) | 3.77 | 12.72 | 2.60 | 11.17 | 3.05 | 10.56 | 5.62 | 15.35 |
| *Djougou (Benin)* | | | | | | | | |
| **Average ($\mu$g.m$^{-3}$)** | **38.25** | | **41.31** | | **37.71** | | **35.70** | |
| Anthropogenic ($\mu$g.m$^{-3}$ and %) | 4.10 | 10.72 | 4.38 | 10.60 | 4.59 | 12.16 | 3.35 | 9.38 |
| Fire ($\mu$g.m$^{-3}$ and %) | 9.18 | 23.99 | 1.34 | 3.23 | 9.36 | 24.82 | 16.84 | 47.17 |
| Dust ($\mu$g.m$^{-3}$ and %) | 13.55 | 35.43 | 23.73 | 57.44 | 12.05 | 31.95 | 4.83 | 13.52 |
| Biogenic ($\mu$g.m$^{-3}$ and %) | 9.95 | 26.01 | 10.76 | 26.05 | 10.44 | 27.68 | 8.66 | 24.25 |
| Salt ($\mu$g.m$^{-3}$ and %) | 1.47 | 3.85 | 1.10 | 2.67 | 1.28 | 3.39 | 2.03 | 5.68 |
| *Niamey (Niger)* | | | | | | | | |
| **Average ($\mu$g.m$^{-3}$)** | **53.76** | | **71.59** | | **52.56** | | **37.09** | |
| Anthropogenic ($\mu$g.m$^{-3}$ and %) | 4.22 | 7.85 | 4.00 | 5.58 | 5.15 | 9.80 | 3.54 | 9.54 |
| Fire ($\mu$g.m$^{-3}$ and %) | 5.73 | 10.66 | 0.56 | 0.78 | 5.47 | 10.41 | 11.15 | 30.06 |
| Dust ($\mu$g.m$^{-3}$ and %) | 36.86 | 68.57 | 61.02 | 85.24 | 34.34 | 65.35 | 15.14 | 40.83 |
| Biogenic ($\mu$g.m$^{-3}$ and %) | 6.31 | 11.75 | 5.58 | 7.79 | 7.08 | 13.48 | 6.31 | 17.01 |
| Salt ($\mu$g.m$^{-3}$ and %) | 0.63 | 1.18 | 0.44 | 0.61 | 0.51 | 0.96 | 0.95 | 2.56 |

**Table 3.** *PM$_{2.5}$ ($\mu$g.m$^{-3}$) average and relative contributions (%) of type of pollution source (anthropogenic, fire, dust, biogenic, sea salt at Cotonou (Benin), Djougou (Benin) and Niamey (Niger). The time averaged periods correspond to each month of May, June July and to the whole period (from May to July).*