# Peer review of "Interactions of Atmospheric Gases and Aerosols with the Monsoon Dynamics over the Sudano-Guinean region during AMMA"

_Atmospheric Chemistry and Physics, 2017_

## Referee Comment (RC1) · Anonymous Referee #1 · 7 Aug 2017

**Deroubaix et al., 2017, ACP, Interactions of Atmospheric Gases and Aerosols with the Monsoon Dynamics over the Sudano-Guinean region during AMMA**

**General Description of manuscript:**
The authors use observations from the West Africa AMMA aircraft campaign in 2006 and an atmospheric chemistry model to diagnose the transport patterns and contributing sources to enhancements in carbon monoxide and PM$_{2.5}$ along a latitudinal transect from the Gulf of Guinea to the Sahel.

**General Comments:**
As presented currently the study appears anecdotal. It is not apparent that the features observed along a very limited longitudinal domain in West Africa apply to the rest of West Africa and to other years. Pease clarify whether the findings in this study are generally applicable to the rest of West Africa and other years? If so, what do the outcomes from this study mean for past/present/future atmospheric composition or development of air quality and/or climate policy?

Why not also compare the model to other parameters measured during the AMMA campaign to assist in interpreting transport patterns and contributing sources and diagnosing what causes differences between modeled and observed PM$_{2.5}$ and CO? These could include measure components of PM$_{2.5}$ (sulfate, ammonium, organic aerosol, nitrate), and CO precursor VOCs, for example.

**Specific Comments:**
p. 2, Lines 17-18: The authors point to economic growth as a driver of emissions from industries, including gas flaring, but the reference they site does not mention economic growth as a driver.

p. 2, Line 19: Do the authors mean "air quality standards" or air quality guidelines? If from WHO these should be guidelines.

p. 3, lines 9-12: Presumably DACCIWA will also contribute to understanding the change in atmospheric composition due to increases in emissions over a rapidly growing region?

p. 3, line 31: Is a 1 month spin-up sufficient for carbon monoxide output from a model, when CO has a lifetime of ~2 months?

p. 5, lines 22-23: Please point out the features that are similar to the Flaounas et al. (2010).

p. 6, line 14: Remove parentheses around the AERONET URL.

p. 6, line 15: Space between number and units (400 nm instead of "400nm").

p. 6, line 15: Provide units for "440-870".

p. 10, line 17: "analyzes" should be analysis.

p. 10, lines 26-27: Point out in Figure 5 the feature that indicates the arrival of the cold tongue.

p. 12, line 31: Fix units.

p. 14, line 11: "perturbated" should be perturbed.

p. 15, line 14: What does "Figure ??" refer to? Is this Figure 11? Indicate on the figure the convective cell.

p. 15, line 5 (bottom of page): Dust is repeated.

p. 16, lines 29-30: There is no context for why the results in this work will be compared to DACCIWA. What new insights will be gained from this comparison that justify mentioning it here?

Figures:
Figure 3:
- What are the statistics in the first and third panel? Does this compare the modeled component to total AOD from the measurements? What's the value in showing this? Why not just compare total modeled and observed AOD?

- The label for the modeled AOD components is confusing. The label is "AOD Anthr." and "AOD Fires", but shouldn't is rather be biomass burning and all other components for clarity? The figure caption suggests this is what is shown.

---

## Referee Comment (RC2) · Anonymous Referee #2 · 10 Aug 2017

General Comments:

This paper looks at the contribution of different emissions sources to CO and PM2.5 over W. Africa and the meteorological conditions that influence the transport of the pollutants within this region. It is a model study using a WRF coupled to CHIMERE and evaluated using measurements made during the AMMA project in 2006. It is the impact of the meteorological conditions on pollution concentrations particularly on the coastal region where the majority of people live that provides novel insight that is worthy of publication. I do have a few concerns that I would like to see addressed before publication.

[Figure]

Specific Comments:

It is not clear to me why in section 3.3, the model meridional simulation of CO and PM2.5 is evaluated against just two flights the on consecutive days (13-14 June), during which an MCS passes through the area (section 3.3.1). On the back of this the model is then used to quantify the modelled pollution source apportionment on a monthly basis. Why not evaluate the model over the whole month? There were flights on other days and with other aircraft. There are several papers published from the AMMA campaign that could have been used to help with this evaluation – see Reeves et al (2010) (www.atmos-chem-phys.net/10/7575/2010/) that gives an overview of the chemical and aerosol characterisation and references therein. Satellite data could also be used for the evaluation. This would also help to evaluate the transport of biomass burning plumes into the region from south of the aircraft flight tracks.

How is the source apportionment (Section 3.3.1) determined? Are separate CO tracers used for the background, anthropogenic and biomass burning? How is this done for PM2.5, in particular considering how the aerosol scheme works? How is the formation of SOA considered? What about mixed aerosols? One of the main scientific questions addressed in this paper is the contribution of different sources to pollutant concentrations, so it is essential that a clear description of the methodology for determining this is included in the paper. Much of the analysis in the paper focuses on anth-PM2.5 so it must be clear how this is defined.

Section 4.22. I really do not understand the conclusions here. In Figure 7, surely a and b are the wrong way around, with the top plot having the 1 a.u. contour at around 7N near Cotonou and the bottom plot having it at 14N near Niamey? Perhaps I am getting muddled by this figure, but it seems extremely odd that the ratio of the coastal to Sahelian tracer is greater in the Sahelian region, especially since the tracer experiment uses arbitrary units and so does not consider the relative strengths of the tracer emissions in each region. The clarity of the discussion could be improved by attention to the English, but I think there is something scientifically wrong here.

[Figure]

P 2, l 22-24: Several values are given for high concentrations of pollutants in these 3 lines. It would be helpful to give the time averages over which these measurements were made as the context of this paragraph is to compare them with the air quality standards which are for specified periods of exposure.

P 4, l 24-25: I'd like to see more details on how WRF is coupled to CHIMERE. Is the CHIMERE transport used or just the chemistry? Time steps for physical processes and chemistry?

P 5, l 24: It would be good to show plots of the TRMM and GPCP data to demonstrate the good agreement.

P 6, l 3-8: Looking at Fig. 2, it seems to me that the precipitation is focused at 2N through much of June and that it is only until mid-late June that it shifts to more to 5N. This is not consistent with the text that says the pre-onset occurs in May.

P 9, l 30: The high CO concentrations at the coast are not so continuous in late June and July.

P 10, l 5-8: Please explain more clearly how precipitation/convection impacts surface CO concentrations. How does it affect the vertical distribution?

P 10, l 15-16: A can't make out any great difference between the pattern at 12N and 13N.

P 12, l 27: Are diurnal patterns included in the emission inventories used?

P 12, l 29-31: In Fig. 8 some of the pollution over the sea on the 10-11 June in the Hovmuller plot appears to progress northwards with time (i.e. bottom left towards top right) rather than be transported out to sea from the land. Fig. 9 suggests it may be coming from other cities further to the south.

P 13, l 33: How can you be sure that the plumes are "overlaying" and not mixed?

The English needs to be improved. I have listed some places where the understanding

is not clear or incorrect because of the English, but there are many minor corrections that need to be made (e.g. the appropriate use of "the" and "a", use of singular and plural) that I have not listed. Technical Comments:

Ensure the initial letters of "Guinean Gulf" are in uppercase, here and throughout the paper.

P 1, l 5-6: It needs to be clear what the 38% relates to. 38% of PM.2.5?

P 1, l 9-10: It is not clear. Are the pollutants emitted near the coast concentrated in the Sahel?

P 1, l 11: "Refining the analysis" – reword the English. Suggest "Focusing the analysis"

P 1, l 13: "overlay" each other?

P 1, l 13: "high pollution level" is ambiguous. High concentrations? High attitude?

P 2, l 2: "washout the atmosphere" change to "wash pollutants out of the atmosphere"

P 2, l 4: "air pollution". Are natural components of the atmosphere pollutants? E.g. Sea salt aerosols?

P 2, l 7: "in megacities" should be "from megacities".

P 2, l 31: "since the last decade". The English is not clear. Do you mean "since" or "during". Note that the main AMMA campaign was in 2006, i.e. more than a decade ago.

P 3, l 13: "This article is dedicated to the pollutants transport over the Guinean Gulf coastal region and focuses on two major pollutant concentrations:". Reword the English, "This article focuses on transport of pollurants over the Guinean Gulf coastal region, in particular on:"

P 3, l 15: Replace "have both an" with "both have a".

P 3, l 17: Replace "pollutants in the" with "pollutants to the".

P 3, l 23: Replace "refines spatially" with "focuses on".

P 3, l 27: It would be useful to provide a figure showing the 2 nested domains.

P 4, l 6: Replace "hourly interpolated" with "interpolated hourly".

P 4, l 8: "better" than what?

P 4, l 27-28: Replace "The anthropogenic emissions are estimated using the HTAP v2 (Hemispheric Transport of Air Pollution) annual totals for the year 2010 by the EDGAR Team," with "The anthropogenic emissions are estimated by the EDGAR Team using the HTAP v2 (Hemispheric Transport of Air Pollution) annual totals for the year 2010,"

P 4, l 31-32: "Taking into account vegetation fires emission fluxes is of primary importance to simulate West African pollution (Giglio et al., 2006)."- The English needs improving.

P 4, l 31 – P5, l2: Be consistent with terms and their combinations: "fire", "vegetation", "biomass burning". How were the two parts split?

P 5, l 4: "a new global soil and surface datasets" – singular or plural?

P 5, l 4: "a satellite-derived aeolian roughness length data" – singular or plural?

P 5, l 14-15: "first analyze and quantify the temporal variability of the pollutants concentrations modeled in the urbanized areas along the Guinean Gulf coast during the whole AMMA-SOP1 period (Redelsperger et al., 2006). First" There are two "firsts" which is confusing. It states that the temporal variability of the pollutants will be analysed, but in the rest of the paragraph I can only see mention of AOD. What about any other pollutants?

P 5, l 29: Replace "interactions constituted" by "interactions, which are made up".

P 6, l 27: Replace "lead" with "leads".

P 7, l 5-6: English needs improving.

P 7, l 28: What is meant by "a South-North gradient is expected moving closer to the Sahara"?

P 7, l 31 and P8, l 4: What is meant by "a gap of concentration"?

P 7, l 7: To test if the model reproduces the MCS, why not compare modelled meteorological parameters with observed – e.g. satellite precipitation.

P 7, l 8: Replace "not realistic" with "unrealistic".

P 7, l 9: Why "Nevertheless"?

P 8, l 31: Presumably by "vegetation emissions" you mean biomass burning rather than biogenic. This needs to be clear and correct throughout the paper.

P 8, l 33-34 and P 9, l 5 and 10: Units of ppb should be microg / m-3.

P 10, l 10: Why "more important"?

P 10, l 11: What "increase"? The English in this sentence needs improving. Consider splitting it in two.

P 10, l 13: A frequency has units of 1/time. Do you mean a periodicity close to 2 weeks?

P 10, l 14: A wouldn't call these features plumes as they are a characteristic of a Hovmuller plot rather than a pollution plume.

P 11, l 19: anti-clockwise?

P 12, l 25: Replace "derives" with "are transported".

P 14, l 11-12: Low and high anthropogenic pollution where? Contonou?

P 15, l 14: "Figure ??", 11.

Fig. 2. Caption "regional domain" should be replaced by "regional model domain".

Fig. 3. The legend only says "anthr" but the captions says "anthropogenic, biogenic and mineral dust". Is the red line the sum of "anthropogenic, biogenic and mineral dust" and "biomass burning emissions"?

Fig. 6 caption: what PM2.5 mass density is the light orange line meant to be? Shading should be lines.

Fig. 7 caption: Shading should be lines. It is not clear if one line is white or both are yellow.

---

## Author Response (AR1)

**Interactions of Atmospheric Gases and Aerosols with the Monsoon Dynamics over the Sudano-Guinean region during AMMA**

Adrien DEROUBAIX, Cyrille FLAMANT, Laurent MENUT, Guillaume SIOUR, Sylvain MAILLER, Solène TURQUETY, Régis BRIANT, Dmitry KHVOROSTYANOV and Suzanne CRUMEYROLLE

Dear Editor,

We thank you and the reviewers for their positive comments and opinions. About their comments, we propose in the following some answers and corrections of the manuscript.

**1 Report 1**

**1.1 General Description of manuscript and General Comments**

The authors use observations from the West Africa AMMA aircraft campaign in 2006 and an atmospheric chemistry model to diagnose the transport patterns and contributing sources to enhancements in carbon monoxide and PM 2.5 along a latitudinal transect from the Gulf of Guinea to the Sahel.

As presented currently the study appears anecdotal. It is not apparent that the features observed along a very limited longitudinal domain in West Africa apply to the rest of West Africa and to other years. Please clarify whether the findings in this study are generally applicable to the rest of West Africa and other years? If so, what do the outcomes from this study mean for past/present/future atmospheric composition or development of air quality and/or climate policy?

The present analysis is meant to be a case study for the year 2006, making use of the wealth of observations acquired during the AMMA programme. The interannual variability has not been investigated and is out of the scope of this paper. We revisited the AMMA 2006 observations to focus on the transport patterns and sources of carbon monoxide and PM2.5 over Southern West Africa, a region that has received less attention than the Sahel during AMMA.

We suspect that some of the features observed occur year after year over Southern West Africa monsoon but we cannot extrapolate from the results of our study. Here, we proposed a new approach focusing on two major pollutants (Carbon monoxide, CO, and fine atmospheric particulate matter, PM2.5) that can be transported far from the sources due to their long lifetime. They are certainly of paramount importance for air quality and climate policies development, but we feel that additional simulations are needed to address this, which is also beyond the scope of the paper.

Why not also compare the model to other parameters measured during the AMMA campaign to assist in interpreting transport patterns and contributing sources and diagnosing what causes differences between modeled and observed PM 2.5 and CO? These could include measure components of PM 2.5 (sulfate, ammonium, organic aerosol, nitrate), and CO precursor VOCs, for example.

We decided to focus on these two important atmospheric components (Carbon monoxide, CO, and fine atmospheric particulate matter, PM2.5) because the data were available. Aerosol Mass Spectrometer data were not available. For CO, we assume that we modeled the two main sources (i.e. Anthropogenic and biomass burning). Given the amount of VOCs, i.e. $> 15$ ppb according to Ancellet et al. (2011), VOCs oxidation must be very low (a few ppb).

**1.2 Specific Comments**

We thank the reviewer for taking the time to go through the spelling and grammar of the manuscript. All proposed corrections have been taken into account in the revised version of the manuscript. Below, we provide answers to the more science oriented questions the referee has.

p. 2, Lines 17-18: The authors point to economic growth as a driver of emissions from industries, including gas flaring, but the reference they site does not mention economic growth as a driver.

The sentence has been changed: *'However, the economic growth over the region drives up anthropogenic emissions: the increase of industries including gas flaring (Asuoha and Osu, 2015), of local fuel-wood burning*

*for stoves and of traffic (Liousse et al., 2010; Hadji et al., 2012; Liousse et al., 2014) with more two-wheel vehicles using very poor fuel quality used (Ndoke and Jimoh, 2005; Assamoi and Liousse, 2010), which are suspected to quickly worsen the air quality.'*

p. 2, Line 19: Do the authors mean 'air quality standards' or air quality guidelines? If from WHO these should be guidelines.

Corrected

p. 3, lines 9-12: Presumably DACCIWA will also contribute to understanding the change in atmospheric composition due to increases in emissions over a rapidly growing region?

The sentence has been changed: *It will contribute to understanding the change in atmospheric composition due to increases in emissions over a rapidly growing region as well as the development of the next generation of accurate models to forecast weather and pollution in southern West Africa (Knippertz et al., 2015).*

p. 3, line 31: Is a 1 month spin-up sufficient for carbon monoxide output from a model, when CO has a lifetime of 2 months?

As for all chemical species, and the principle of an area limited domain model, the lifetime of the species is not a constraint. The 'aged' concentrations are already modeled with the global climatological model, providing hourly boundary conditions. These concentrations are injected into our regional model depending on the wind direction and speed. The 'fresh' concentrations are explicitly hourly emitted in the domain.

The real constraint motivating the use of a spin-up time is the transport of the species into the domain: we want to ensure that for the first modeled hour, all possible species, due to a previous transport, are well present in the domain. There is no link to the lifetime, but depends on the transport and the domain size only.

p. 5, lines 22-23: Please point out the features that are similar to the Flaounas et al. (2010).

The paragraph related with Flaounas et al. (2010) has been modified such as: *'During this period, the precipitation location and rate will play a crucial role on the modeled surface $PM_{2.5}$ concentrations. As a validation for this variable, the methodology of Flaounas et al. (2010) is used: precipitation rates are averaged between 8.5W and 8.5E. Day-to-day variability is smoothed by applying a moving average of $\pm 2$ days. Figure 2 is directly comparable to the Flaounas et al. (2010) study using the same period and averaged region. In May and June, observed and modeled precipitations occur mainly over the ocean (below 5N). From late June on, the main precipitation areas move over the continent (above 5N) and reach the Sahel (at about 13N). Figure 2 shows that the modeled precipitation spatial patterns are in good agreement with the two satellite observations (TRMM and GPCP) presented in their study (see Figure 3 of Flaounas et al. (2010)). '*

p. 6, line 14: Remove parentheses around the AERONET URL.
p. 6, line 15: Space between number and units (400 nm instead of '400nm').
p. 6, line 15: Provide units for '440-870'.
p. 10, line 17: 'analyzes' should be analysis.

All four points have been corrected.

p. 10, lines 26-27: Point out in Figure 5 the feature that indicates the arrival of the cold tongue.

This sentence has been revised: *At the end of the period, when precipitation occurs inland and anth-$PM_{2.5}$ is low, the meteorological situation changes suddenly over the ocean showing the cold tongue arrival located at the Equator, which is associated with increased wind speed between the Equator and the coast, as detailed by Meynadier et al. (2016).*

p. 12, line 31: Fix units.
p. 14, line 11: 'perturbated' should be perturbed.
p. 15, line 5 (bottom of page): Dust is repeated.

All three points have been corrected.

p. 15, line 14: What does 'Figure ??' refer to? Is this Figure 11? Indicate on the figure the convective cell.

Thanks for picking this up. Yes indeed, we meant Figure 12 (previously 11). A red ellipse has been superimposed on Figure 12 to indicate the location of the convective cell.

p. 16, lines 29-30: There is no context for why the results in this work will be compared to DACCIWA. What new insights will be gained from this comparison that justify mentioning it here?

This sentence has been changed: *Concerning air quality and climate policy development, we have shown that the export of anthropogenic pollutant from the Guinean coast toward the North could lead to cross boundary pollution plumes. This result will be confirmed by comparing to the 2016 DACCIWA campaign observations in order to propose strategy to reduce the atmospheric pollution in West Africa.*

Figures:
Figure 3: What are the statistics in the first and third panel? Does this compare the modeled component to total AOD from the measurements? Whats the value in showing this? Why not just compare total modeled and observed AOD? The label for the modeled AOD components is confusing. The label is 'AOD Anthr.' and 'AOD Fires', but shouldnt is rather be biomass burning and all other components for clarity? The figure caption suggests this is what is shown.

There is no value to keep the statistics because the scores are very slightly improved by adding biomass burning emission. It has been removed. Concerning the legend, the labels ('AOD Anthr.' and 'AOD Fires') have been replaced (by 'with biomass burning' and 'without biomass burning'). The new figure is presented below ( Figure 1 of this document).
Moreover, a detailed explanation of how the different contributions are obtained has been added in Section 2.2.

**2 Report 2**

**2.1 General Comments**

This paper looks at the contribution of different emissions sources to CO and PM2.5 over W. Africa and the meteorological conditions that influence the transport of the pollutants within this region. It is a model study using a WRF coupled to CHIMERE and evaluated using measurements made during the AMMA project in 2006. It is the impact of the meteorological conditions on pollution concentrations particularly on the coastal region where the majority of people live that provides novel insight that is worthy of publication. I do have a few concerns that I would like to see addressed before publication.

**2.2 Specific Comments**

It is not clear to me why in section 3.3, the model meridional simulation of CO and PM2.5 is evaluated against just two flights the on consecutive days (13-14 June), during which an MCS passes through the area (section 3.3.1). On the back of this the model is then used to quantify the modelled pollution source apportionment on a monthly basis. Why not evaluate the model over the whole month? There were flights on other days and with other aircraft. There are several papers published from the AMMA campaign that could have been used to help with this evaluation  see Reeves et al (2010) (www.atmos-chem-phys.net/10/7575/2010/) that gives an overview of the chemical and aerosol characterisation and references therein. Satellite data could also be used for the evaluation. This would also help to evaluate the transport of biomass burning plumes into the region from south of the aircraft flight tracks.

The section 3.3 deals with modeled and observed CO and PM2.5 during two meridional flight trajectories on consecutive days (13-14 June) because it provides data from the coast to the Sahel, thus it fits exactly the scope of the study focusing on the Sudano-Guinean region. New insights are presented gained from the AMMA 2006 observations as we focus on the transport patterns and sources of CO and PM2.5 over the Sudano-Guinean region.

[Figure]

Figure 1: *Observed daily averages of AERONET level 2 AOD and Angström exponent (black dots) at Djougou (Benin) and Banizoumbou compared to the modeled time series with a splitting to extract the relative contribution between without biomass burning emissions (including anthropogenic, biogenic, sea salt and mineral dust; all four in blue) and with biomass burning emissions (in red).*

Moreover, it has been difficult to work with validated data from the AMMA database. Many datasets used in preliminary evaluations were partly unusable after contacting responsible people of the dataset. We have contacted FAAM team (Graeme NOTT) for data from the BAe 146 of the 20-21 July but it was not possible to produce PM concentration because: 'during this period no calibration of the PCASP was carried out by FAAM' and in the FORTRAN code, a mistake has been done using constant flow rate. Suzanne CRUMEYROLLE has provided the only aircraft data that we are confident but aerosol speciation was not available. The two flights used in our study are meridional transects that fit exactly with the purpose of the study.

How is the source apportionment (Section 3.3.1) determined? Are separate CO tracers used for the background, anthropogenic and biomass burning? How is this done for PM2.5, in particular considering how the aerosol scheme works? How is the formation of SOA considered? What about mixed aerosols? One of the main scientific questions addressed in this paper is the contribution of different sources to pollutant concentrations, so it is essential that a clear description of the methodology for determining this is included in the paper. Much of the analysis in the paper focuses on anth-PM2.5 so it must be clear how this is defined.

References describing how the aerosol scheme works in the CHIMERE model were missing in the manuscript. In the revised version of the manuscript, we now describe it as well as how the source apportionment is determined for CO and for PM2.5, which has been added in Section 2.2:
*'Menut et al. (2016) have detailed and analyzed aerosol speciation and size distribution in the CHIMERE model during the summer 2013 over Europe and Africa using the AERONET network for AOD and EMEP network for PM concentrations. For the AOD calculation, the aerosol optical scheme in the CHIMERE model considers mixed aerosols following the 'core-shell' hypothesis detailed in Péré et al. (2009) and evaluated in Péré et al. (2010).*
*In order to quantify the PM2.5 source apportionment, we assume that it is possible to split aerosols in different families depending on the sources because their chemical compositions are different: Mineral, Biogenic, Salt and Anthropogenic. Given that anthropogenic and biomass burning aerosols have similar compositions, we have done two simulations with and without biomass burning emissions to split their contributions. The gas phase chemical scheme for SOA formation explained in Bessagnet et al. (2010) takes into account three anthropogenic and three biogenic hydrophilic species, three hydrophobic species with different saturations, and two surrogate compounds for the isoprene oxidation products.*
*The source apportionment has been determined for CO considering three main contributors (anthropogenic sources, biomass burning sources and long-range transport). Consequently, three simulations have been done: one without any emission source in the domain for the background concentration, one with the anthropogenic emission only, and a last one with the anthropogenic and biomass burning emissions.'*

Section 4.22. I really do not understand the conclusions here. In Figure 7, surely a and b are the wrong way around, with the top plot having the 1 a.u. contour at around 7N near Cotonou and the bottom plot having it at 14N near Niamey? Perhaps I am getting muddled by this figure, but it seems extremely odd that the ratio of the coastal to Sahelian tracer is greater in the Sahelian region, especially since the tracer experiment uses arbitrary units and so does not consider the relative strengths of the tracer emissions in each region. The clarity of the discussion could be improved by attention to the English, but I think there is something scientifically wrong here.

We thank the reviewer for reporting this mistake. In this Figure (now Figure 8) , a) and b) were the wrong way around. This has been corrected.
Real anthropogenic emissions represent the amount of what each city emits: their magnitude is thus representative of each location, the activity sectors and the population. After emissions, pollutants are transformed by transport, mixing, deposition and chemistry. In this case, and if we want a realistic estimate of a concentration far from the sources, it is necessary to have emissions really representative of the size of each urbanized area. The problem is different with the tracers: we don't want to have a realistic value, but just to know the percentage of what arrived at a remote location. Thus, we need to emit the same amount at every location to have the exact percentage at the studied remote location. This enables to quantify that in Niamey, Cotonou tracer concentration is about 9% (of the 1 a.u. isocontour presented in Figure 8-a), while in Cotonou, Niamey tracer concentration is about 0.03% (of the 1 a.u. isocontour presented in Figure 8-b).
In Section 4.2.2, a sentence has been added for clarity: *'The tracer experiment uses arbitrary units and considers the same quantity of tracers emitted in each town.'*

The second paragraph of this section has been changed: *'Tracers emitted at the coast indicate that there is an important transport of coastal pollutants toward the North in the PBL. On the other hand, there is no significant transport of tracers emitted in the Sahel toward the coast. In Niamey, Cotonou tracer concentration is about 9% (of the 1 a.u. isocontour presented in Figure 8-a), while in Cotonou, Niamey tracer concentration is about 0.03% (of the 1 a.u. isocontour presented in Figure 8-b). In the HTAP anthropogenic inventories (presented in Figure 1), the anth-$PM_{2.5}$ (respectively anth-CO) is $\approx$ 103 (735) $kg.km^{-2}.day^{-1}$ in Niamey and $\approx$ 438 (7707) $kg.km^{-2}.day^{-1}$ in Cotonou. Therefore, an important part of the pollution over the Sahel has been emitted at the coast and it contributes to a maximum of anthropogenic pollution in June over the Sahel. In conclusion, the high concentration over the Sahel is due to the existence of a meridional atmospheric cell, which acts at accumulating pollutants emitted locally and remotely at the coast.'*

P 2, l 22-24: Several values are given for high concentrations of pollutants in these 3 lines. It would be helpful to give the time averages over which these measurements were made as the context of this paragraph is to compare them with the air quality standards which are for specified periods of exposure.

Time averages have been added: *'based on half-hour averages'* for Baumbach et al. (1995); *'based on 1-min averages'* for Dionisio et al. (2010); *'based on daily averages'* for Boman et al. (2009)

P 4, l 24-25: Id like to see more details on how WRF is coupled to CHIMERE. Is the CHIMERE transport used or just the chemistry? Time steps for physical processes and chemistry?

In this study, we present offline simulations (CHIMERE is forced by WRF). For chemistry and aerosol simulations, the concentrations are calculated using the chemistry, transport, mixing and deposition equations implemented in the CHIMERE model. For clarity, two sentences have been added: *'The WRF and CHIMERE models are run offline on the same horizontal grids for the continental and regional domains'* and *'The time step is set to 10 minutes for the physical processes and 5 minutes for the chemistry, which could change depending on the Courant-Friedrichs-Lewy condition.'*

P 5, l 24: It would be good to show plots of the TRMM and GPCP data to demonstrate the good agreement.

These two plots have been presented by Flaounas et al. (2010) as well as the comparison to the different WRF parametrization. We do not aim at focusing on the precipitation patterns in the present study. We have specifically pointed out comparable patterns of our Figure 2 and Figure 3 of Flaounas et al. (2010). The first paragraph of Section 3.1 has been modified such as: *'During this period, the precipitation location and rate will play a crucial role on the modeled surface $PM_{2.5}$ concentrations. As a validation for this variable, the methodology of Flaounas et al. (2010) is used: precipitation rates are averaged between 8.5W and 8.5E. Day-to-day variability is smoothed by applying a moving average of $\pm 2$ days. Figure 2 is directly comparable to the Flaounas et al. (2010) study using the same period and averaged region. In May and June, observed and modeled precipitations occur mainly over the ocean (below 5N). From late June on, the main precipitation areas move over the continent (above 5N) and reach the Sahel (at about 13N). Figure 2 shows that the modeled precipitation spatial patterns are in good agreement with the two satellite observations (TRMM and GPCP) presented in their study (see Figure 3 of Flaounas et al. (2010)). '*

P 6, l 3-8: Looking at Fig. 2, it seems to me that the precipitation is focused at 2N through much of June and that it is only until mid-late June that it shifts to more to 5N. This is not consistent with the text that says the pre-onset occurs in May.

The section 3.1 has been modified to be consistent with the figure and the three periods defined (see above).

P 9, l 30: The high CO concentrations at the coast are not so continuous in late June and July.

*'during the whole period'* has been changed to *'from the beginning of May to late June'*

P 10, l 5-8: Please explain more clearly how precipitation/convection impacts surface CO concentrations. How does it affect the vertical distribution?

In this section, we identify the changes which are analyzed in the following. The sentence: *'In July, the variability is mostly consistent with precipitation rates after the onset, which suggest that surface versus vertical distribution has changed by the convection associated with large scale precipitation.'* has been replaced by *'In July, the variability is mostly consistent with precipitation rates after the onset, suggesting modifications of transport and deposition patterns by the convection associated with large scale precipitation.'*

P 10, l 15-16: A cant make out any great difference between the pattern at 12N and 13N.

*'at 12N'* has been removed. This sentence was unclear because we were not comparing 12N and 13N. This paragraph has been modified (see above).

P 12, l 27: Are diurnal patterns included in the emission inventories used?

*'when the convection and NLLJ are weak'* has been removed because diurnal patterns of anthropogenic pollution are mostly driven by diurnal variation of the emissions.

P 12, l 29-31: In Fig. 8 some of the pollution over the sea on the 10-11 June in the Hovmuller plot appears to progress northwards with time (i.e. bottom left towards top right) rather than be transported out to sea from the land. Fig. 9 suggests it may be coming from other cities further to the south.

This paragraph has been modified such as: *'There is a transition from low to high concentration of anthropogenic pollution from 8 to 12 June. Anthropogenic pollution is modeled over the sea from 10 to 11 June. It is interesting to note that precipitation occurs inland on 11 June (between 18 UTC and 00 UTC), then high modeled concentrations persist during the night of 11-12 June. This precipitation event reflects a change in the wind patterns, which induces a change in the transport of pollutants, leading to surface concentrations up to 8 $\mu g.m^{-3}$ in Cotonou.'*

P 13, l 33: How can you be sure that the plumes are 'overlaying' and not mixed?

You are right, it is mixed. We are analyzing the surface level. If the tracers associated to the different cities are at the same location, it means that the pollution from the different cities is mixed.

The English needs to be improved. I have listed some places where the understanding is not clear or incorrect because of the English, but there are many minor corrections that need to be made (e.g. the appropriate use of 'the' and 'a', use of singular and plural) that I have not listed.

**2.3   Technical Comments**

Ensure the initial letters of 'Guinean Gulf' are in uppercase, here and throughout the paper.

OK

P 1, l 5-6: It needs to be clear what the 38% relates to. 38% of PM.2.5?

This sentence has been modified from *'For $PM_{2.5}$, desert dust decreases from $\approx$ 38 % in May to $\approx$ 5 % in July;'* to *'Desert dust decreases from $\approx$ 38 % in May to $\approx$ 5 % in July of $PM_{2.5}$ concentration'*

P 1, l 9-10: It is not clear. Are the pollutants emitted near the coast concentrated in the Sahel?

In order to be clear, this sentence has been split in two sentences: *'Air masses dynamics concentrate pollutants emitted in the Sahel due to a meridional atmospheric cell. Moreover a part of the pollution emitted remotely at the coast is transported and accumulated over the Sahel.'*

P 1, l 11: 'Refining the analysis' reword the English. Suggest 'Focusing the analysis'

OK

P 1, l 13: 'overlay' each other?

They are mixed and not overlaying. Corrected

P 1, l 13: 'high pollution level' is ambiguous. High concentrations? High attitude?

Indeed 'level is ambiguous. It should be either 'concentration' or 'altitude'. The manuscript has been entirely modified to remove this ambiguity.

P 2, l 2: 'washout the atmosphere' change to 'wash pollutants out of the atmosphere'

OK

P 2, l 4: 'air pollution'. Are natural components of the atmosphere pollutants? E.g. Sea salt aerosols?

'air pollution sources' has been replaced by 'aerosol and gas sources'

P 2, l 7: 'in megacities' should be 'from megacities'.

OK

P 2, l 31: 'since the last decade'. The English is not clear. Do you mean 'since' or 'during'. Note that the main AMMA campaign was in 2006, i.e. more than a decade ago.

'since the last decade' has been deleted.

P 3, l 13: 'This article is dedicated to the pollutants transport over the Guinean Gulf coastal region and focuses on two major pollutant concentrations:'. Reword the English, 'This article focuses on transport of pollutants over the Guinean Gulf coastal region, in particular on:'

OK

P 3, l 15: Replace 'have both an' with 'both have a'.
P 3, l 17: Replace 'pollutants in the' with 'pollutants to the'.
P 3, l 23: Replace 'refines spatially' with 'focuses on'.

All last three points have been corrected.

P 3, l 27: It would be useful to provide a figure showing the 2 nested domains.

The latitudes and longitudes of the two domains are given in the beginning of Section 2. The results of the article concern only the West African region. Furthermore, we do not compare the coarse and fine resolutions. It is why we want to focus only on the regional domain (presented in Figure 1).

P 4, l 6: Replace 'hourly interpolated' with 'interpolated hourly'.

OK

P 4, l 8: 'better' than what?

The sentence has been changed and 'better' removed.

P 4, l 27-28: Replace 'The anthropogenic emissions are estimated using the HTAP v2 (Hemispheric Transport of Air Pollution) annual totals for the year 2010 by the EDGAR Team,' with 'The anthropogenic emissions are estimated by the EDGAR Team using the HTAP v2 (Hemispheric Transport of Air Pollution) annual totals for the year 2010,'

OK

P 4, l 31-32: 'Taking into account vegetation fires emission fluxes is of primary importance to simulate West African pollution (Giglio et al., 2006).'- The English needs improving.

This sentence has been modified: 'Biomass burning emission from Central Africa is of primary importance to simulate West African pollution (Giglio et al., 2006)'

P 4, l 31  P5, l2: Be consistent with terms and their combinations: 'fire', 'vegetation', 'biomass burning'. How were the two parts split?

The manuscript has been revised to use only 'biomass burning'.
Since the incomplete combustion is both included in anthropogenic inventories (local urban burning) and forests biomass burning inventories, the simulation was designed to split these two parts. It is now explained in Section 2.2

P 5, l 4: 'a new global soil and surface datasets'  singular or plural?
P 5, l 4: 'a satellite-derived aeolian roughness length data'  singular or plural?

This is singular because it is a unique dataset composed of two satellite retrievals.
The sentence has been modified such as: 'The mineral dust sources are obtained using the GARLAP (Global Aeolian Roughness Lengths from ASCAT and PARASOL) new global soil and surface dataset made from satellite-derived aeolian roughness lengths with a 6 km spatial resolution, as detailed in Mailler et al. (2016).'

P 5, l 14-15: 'first analyze and quantify the temporal variability of the pollutants concentrations modeled in the urbanized areas along the Guinean Gulf coast during the whole AMMA-SOP1 period (Redelsperger et al.,

2006). First' There are two 'firsts' which is confusing. It states that the temporal variability of the pollutants will be analysed, but in the rest of the paragraph I can only see mention of AOD. What about any other pollutants?

*This sentence has been modified: 'In this section, we analyze the temporal variability of precipitation, gas and aerosol during the whole AMMA-SOP1 period (Redelsperger et al., 2006).'*

P 5, l 29: Replace 'interactions constituted' by 'interactions, which are made up'.

OK

P 6, l 27: Replace 'lead' with 'leads'.

OK

P 7, l 5-6: English needs improving.

*'Two flights made during a 'North-South land-atmosphere-ocean interaction' mission plans have been conducted over the Cotonou-Niamey meridional transect on 13 and 14 June 2006.' has been modified such as: 'We are studying two flights conducted along a meridional transect between Cotonou and Niamey on 13 and 14 June 2006 as part of a 'North-South land-atmosphere-ocean interaction' survey mission.'*

P 7, l 28: What is meant by 'a South-North gradient is expected moving closer to the Sahara'?

*'a South-North gradient is expected moving closer to the Sahara' is unclear and it has been modified such as: 'For PM$_{2.5}$, a South-North gradient is expected with the highest concentrations close to the Sahara.'*

P 7, l 31 and P8, l 4: What is meant by 'a gap of concentration'?

*This has been replaced by: 'important increase of the concentration'*

P 7, l 7: To test if the model reproduces the MCS, why not compare modelled meteorological parameters with observed e.g. satellite precipitation.

*A new figure has been added in Section 3.3.1 (presented in Figure 2 of this document), which aims at presenting the meteorological situation on 13-14 June as well as validating the model meteorology. The first paragraph of this section has been modified such as: 'We are studying two flights conducted along a meridional transect between Cotonou and Niamey on 13 and 14 June 2006 as part of a 'North-South land-atmosphere-ocean interaction' survey mission. During these two days, the WAM dynamics over the area were perturbed by the presence of a MCS. It developed over the Jos Plateau (Nigeria) around 16:00 UTC, reaching the Benin-Nigeria border at 20:00 UTC and moved southwestward across Benin overnight and into central Ghana, as already described in Flamant et al. (2009) and (Crumeyrolle et al., 2011). The model reproduces the location of this MCS but earlier than in the observations, i.e. reaching the Benin-Nigeria border at 10:00 UTC (Figure 4). The MCS interacts with the dust layer coming from the Sahara (especially from the Bodele depression), changing the dust load and vertical distribution over Benin and Niger. Associated with subsidence in the wake of the MCS, there is a lowering of the dust layer height (Flamant et al., 2009). '*

P 7, l 8: Replace 'not realistic' with 'unrealistic'.

OK

P 7, l 9: Why 'Nevertheless'?

*This sentence has been modified: 'Nevertheless, the order of magnitude of the CO and PM$_{2.5}$ concentrations are good in agreement with observations. '*

P 8, l 31: Presumably by 'vegetation emissions' you mean biomass burning rather than biogenic. This needs to be clear and correct throughout the paper.

OK

P 8, l 33-34 and P 9, l 5 and 10: Units of ppb should be microg / m-3.

OK

P 10, l 10: Why 'more important'?
P 10, l 11: What 'increase'? The English in this sentence needs improving. Consider splitting it in two.

[Figure]

Figure 2: *(top) EUMETSAT visible image of the Cotonou area of the 13 June 2006 at 20 UTC (from NAScube (http://nascube.univ-lille1.fr); (bottom) Map of Cotonou area for the 13 June 2006 at 12 UTC with wind vectors at 10 m (orange arrows), precipitation (blue shading). The two flight trajectories are displayed with the red line for the 13 June and with the green line for the 14 June.*

For the two last points, the sentences: *'The same behavior is observed for the surface concentrations of $PM_{2.5}$. The week to week variability is more important. This increase is probably due to the longer CO lifetime compared with that of PM (being less chemically active and without settling), the CO concentrations are more homogeneously mixed in a large latitudinal area from the coast to more than 16 up to the North.'*, have been reworded: *'The same behavior is observed for the surface concentrations of $PM_{2.5}$. The week to week variability is greater than for anth-CO, which is probably due to the longer lifetime of CO compared with that of PM (being less chemically active and less prone to settling). CO is more homogeneously mixed than PM in a large latitudinal area spanning from the coast to latitudes higher than 16. '*

P 10, l 13: A frequency has units of 1/time. Do you mean a periodicity close to 2 weeks?

OK

P 10, l 14: A wouldnt call these features plumes as they are a characteristic of a Hovmuller plot rather than a pollution plume.

*'latitudinal plumes'* has been replaced by: *'latitudinal patterns'*

P 11, l 19: anti-clockwise?

Yes, it has been corrected.

P 12, l 25: Replace 'derives' with 'are transported'.

OK

P 14, l 11-12: Low and high anthropogenic pollution where? Contonou?

Yes, *'in Cotonou'* has been added.

P 15, l 14: 'Figure ??', 11.

OK

Fig. 2. Caption 'regional domain' should be replaced by 'regional model domain'.

OK

Fig. 3. The legend only says 'anthr' but the captions says 'anthropogenic, biogenic and mineral dust'. Is the red line the sum of 'anthropogenic, biogenic and mineral dust' and 'biomass burning emissions'?

In red, this is anthropogenic plus biogenic plus mineral dust plus sea salt. The legend's labels ('AOD Anthr.' and 'AOD Fires') have been replaced (by 'with biomass burning' and 'without biomass burning').

Fig. 6 caption: what PM2.5 mass density is the light orange line meant to be? Shading should be lines.

We tried to remove shading but it leads to too much contours, which is difficult to read. In the end, we use shading for the meridional wind speed and contours for the anthropogenic PM2.5 concentration. The light orange line has been removed and we focuse only on anthropogenic PM2.5 concentration = 4.0 $\mu$g.m$^{-3}$ (orange) and = 5.0 $\mu$g.m$^{-3}$ (violet). The new figure is presented below ( Figure 3 of this document).

Fig. 7 caption: Shading should be lines. It is not clear if one line is white or both are yellow.

As for the previous figure, we have removed the white line. The comment of this figure is now focused on only two isocontours. The new figure is presented below ( Figure 4 of this document).

[revised manuscript text omitted]

---

## Referee Report (RR1)

**Re-review of Deroubaix et al., 2017, ACP, Interactions of Atmospheric Gases and Aerosols with the Monsoon Dynamics over the Sudano-Guinean region during AMMA**

**General Description of manuscript:**

The authors use observations from the West Africa AMMA aircraft campaign in 2006 and an atmospheric chemistry model to diagnose the transport patterns and contributing sources to enhancements in carbon monoxide and $PM_{2.5}$ along a latitudinal transect from the Gulf of Guinea to the Sahel. The authors have for the most part addressed the comments from reviewers in the first round. There are some remaining concerns that are indicated below that the authors should address before publication in ACP.

**Follow-up Comments:**

Authors: "We decided to focus on these two important atmospheric components (Carbon monoxide, CO, and fine atmospheric particulate matter, PM2.5) because the data were available. Aerosol Mass Spectrometer data were not available. For CO, we assume that we modeled the two main sources (i.e. Anthropogenic and biomass burning). Given the amount of VOCs, i.e. > 15 ppb according to Ancellet et al. (2011), VOCs oxidation must be very low (a few ppb)"

Re-review: There was a Quadrupole Aerosol Mass Spectrometer (Q-AMS) onboard the BAe-146 aircraft (see for example Capes et al., 2009) that includes measurements of fine particulate matter components. These can provide a compelling evaluation of the model and additional information about the composition of aerosols in your study. The VOCs concentration of 15 ppb has the potential to influence CO concentrations in the region, if the number of carbons of each VOC, the reactivity, unmeasured VOCs, measured VOCs not listed in Ancellet et al. (2009) (e.g. methacrolein, methyl ethyl ketone, methyl vinyl ketone, reactive aromatics, and higher order alkanes and alkenes), and the contribution of the non-background sources (67%) are taken into consideration.

Authors: "The sentence has been changed: 'However, the economic growth over the region drives up anthropogenic emissions: the increase of industries including gas flaring (Asuoha and Osu, 2015), of local fuel-wood burning for stoves and of traffic (Liousse et al., 2010; Hadji et al., 2012; Liousse et al., 2014) with more two-wheel vehicles using very poor fuel quality used (Ndoke and Jimoh, 2005; Assamoi and Liousse, 2010), which are suspected to quickly worsen the air quality.'"

Re-review: Thank you for changing the sentence. Unfortunately now it makes no sense. Without the references it reads as follows: 'However, the economic growth over the region drives up anthropogenic emissions: the increase of industries including gas flaring, of local fuel-wood burning for stoves and of traffic with more two-wheel vehicles using very poor fuel quality used, which are suspected to quickly worsen the air quality. Consider breaking up the sentence or rearranging to make the intention clear.

Authors: "As for all chemical species, and the principle of an area limited domain model, the lifetime of the species is not a constraint. The 'aged' concentrations are already modeled with the global climatological model, providing hourly boundary conditions. These concentrations are injected into our regional model depending on the wind direction and speed. The 'fresh' concentrations are explicitly hourly emitted in the domain. The real constraint motivating the use of a spin-up time is the transport of the species into the domain: we want to ensure that for the first modeled hour, all possible species, due to a previous transport, are well present in the domain. There is no link to the lifetime, but depends on the transport and the domain size only."

Re-review: State briefly in the paper why the spin up is conducted so that readers appreciate that it isn't a chemical initialization.

New issue not addressed in the initial review:
Page 3, Lines 18-19: The authors refer to biomass burning as natural. Is it? If so, is there literature to support its categorization?

**Reviewer References:**
Ancellet et al., Atmos. Chem. Phys., 11, 6349–6366, doi:10.5194/acp-11-6349-2011, 2011.
Capes et al., Atmos. Chem. Phys., 9, 3841–3850, doi:10.5194/acp-9-3841-2009, 2009.

---

## Author Response (AR2)

**Interactions of Atmospheric Gases and Aerosols with the Monsoon Dynamics over the Sudano-Guinean region during AMMA**

Adrien DEROUBAIX, Cyrille FLAMANT, Laurent MENUT, Guillaume SIOUR, Sylvain MAILLER, Solène TURQUETY, Régis BRIANT, Dmitry KHVOROSTYANOV and Suzanne CRUMEYROLLE

Dear Editor,

We thank you and the referee for these minor comments. We propose in the following some answers and corrections of the manuscript.

**1 Report 1**

**1.1 General Description of manuscript**

The authors use observations from the West Africa AMMA aircraft campaign in 2006 and an atmospheric chemistry model to diagnose the transport patterns and contributing sources to enhancements in carbon monoxide and PM2.5 along a latitudinal transect from the Gulf of Guinea to the Sahel. The authors have for the most part addressed the comments from reviewers in the first round. There are some remaining concerns that are indicated below that the authors should address before publication in ACP. The authors use observations from the West Africa AMMA aircraft campaign in 2006 and an atmospheric chemistry model to diagnose the transport patterns and contributing sources to enhancements in carbon monoxide and PM2.5 along a latitudinal transect from the Gulf of Guinea to the Sahel.

**1.2 Follow-up Comments:**

**Authors:** We decided to focus on these two important atmospheric components (Carbon monoxide, CO, and fine atmospheric particulate matter, PM2.5) because the data were available. Aerosol Mass Spectrometer data were not available. For CO, we assume that we modeled the two main sources (i.e. Anthropogenic and biomass burning). Given the amount of VOCs, i.e. > 15 ppb according to Ancellet et al. (2011), VOCs oxidation must be very low (a few ppb).

**Re-review:** There was a Quadrupole Aerosol Mass Spectrometer (Q-AMS) onboard the BAe-146 aircraft (see for example Capes et al., 2009) that includes measurements of fine particulate matter components. These can provide a compelling evaluation of the model and additional information about the composition of aerosols in your study. The VOCs concentration of 15 ppb has the potential to influence CO concentrations in the region, if the number of carbons of each VOC, the reactivity, unmeasured VOCs, measured VOCs not listed in Ancellet et al. (2009) (e.g. methacrolein, methyl ethyl ketone, methyl vinyl ketone, reactive aromatics, and higher order alkanes and alkenes), and the contribution of the nonbackground sources (67%) are taken into consideration.

Data from the Q-AMS onboard BAe-146 are not available on the AMMA database (database.amma-international.org). A sentence has been modified in the introduction of Section 3.3: *'In this section, the modeled CO and $PM_{2.5}$ concentrations are compared to aircraft observations collected during the AMMA campaign (Section 3.3.1), available on the AMMA database (albeit other data sets exist).'*

The last sentence of our answer was wrong. We mean that the amount of VOCs is low. Observations onboard UK BAe-146 aircraft has measured VOCs concentration lower than 10 ppb, *e.g.* Capes et al. (2009); Ancellet et al. (2011). Given this low amount, CO produced by VOCs oxidation is low (a few ppb). A sentence has been added in Section 2-2: *'The oxidation of Volatile Organic Carbon gases (VOCs), which leads to CO formation is also taken into account in the anthropogenic sources. However, the amount of VOCs is low, for instance, in BAe-146 measurements VOCs concentration is lower than 10 ppb, e.g. Capes et al. (2009); Ancellet et al. (2011). Thus CO produced by VOCs oxidation is low (a few ppb).'*

**Authors:** The sentence has been changed: 'However, the economic growth over the region drives up anthropogenic emissions: the increase of industries including gas flaring (Asuoha and Osu, 2015), of local fuel-wood burning for stoves and of traffic (Liousse et al., 2010; Hadji et al., 2012; Liousse et al., 2014) with more two-wheel vehicles using very poor fuel quality used (Ndoke and Jimoh, 2005; Assamoi and Liousse, 2010), which are suspected to quickly worsen the air quality.'

**Re-review:** Thank you for changing the sentence. Unfortunately now it makes no sense. Without the references it reads as follows: 'However, the economic growth over the region drives up anthropogenic emissions:

the increase of industries including gas flaring, of local fuel-wood burning for stoves and of traffic with more two-wheel vehicles using very poor fuel quality used, which are suspected to quickly worsen the air quality. Consider breaking up the sentence or rearranging to make the intention clear.'

This sentence has been divided such as: *'However, the economic growth over the region drives up anthropogenic emissions. There are increases of industries including gas flaring (Asuoha and Osu, 2015), of local fuel-wood burning for stoves and of traffic (Liousse et al., 2010; Hadji et al., 2012; Liousse et al., 2014). Moreover, the increase of two-wheel vehicles using very poor fuel quality is suspected to quickly worsen the air quality (Ndoke and Jimoh, 2005; Assamoi and Liousse, 2010).'*

**Authors:** As for all chemical species, and the principle of an area limited domain model, the lifetime of the species is not a constraint. The 'aged' concentrations are already modeled with the global climatological model, providing hourly boundary conditions. These concentrations are injected into our regional model depending on the wind direction and speed. The 'fresh' concentrations are explicitly hourly emitted in the domain. The real constraint motivating the use of a spin-up time is the transport of the species into the domain: we want to ensure that for the first modeled hour, all possible species, due to a previous transport, are well present in the domain. There is no link to the lifetime, but depends on the transport and the domain size only.

**Re-review:** State briefly in the paper why the spin up is conducted so that readers appreciate that it isnt a chemical initialization.

A sentence has been added in the manuscript: *'The spin-up time aims at ensuring that all pollutants emitted outside of the modeled domain are present in the domain (depending on the wind speed and direction) since the first modeled hour.'*

**New issue not addressed in the initial review:** Page 3, Lines 18-19: The authors refer to biomass burning as natural. Is it? If so, is there literature to support its categorization?

We thank the referee for picking this up. Biomass burning is both natural and anthropogenic. This sentence has been changed in the manuscript from *'$PM_{2.5}$ concentrations are dominated by natural sources'* to *'
[revised manuscript text omitted]